# The Blood Circulating Rare Cell Population. What Is It and What Is It Good for?

**DOI:** 10.3390/cells9040790

**Published:** 2020-03-25

**Authors:** Stefan Schreier, Wannapong Triampo

**Affiliations:** 1School of Bioinnovation and Bio-based Product Intelligence, Faculty of Science, Mahidol University, Rama VI Rd, Bangkok 10400, Thailand; stefan@tschiersky.de; 2Thailand Center of Excellence in Physics, Ministry of Higher Education, Science, Research and Innovation, 328 Si Ayutthaya Road, Bangkok 10400, Thailand; 3Department of Physics, Faculty of Science, Mahidol University, Bangkok 10400, Thailand

**Keywords:** circulating rare cells, liquid biopsy, bone marrow-derived progenitor cells, non-hematopoietic cells, cancer, rare cell population

## Abstract

Blood contains a diverse cell population of low concentration hematopoietic as well as non-hematopoietic cells. The majority of such rare cells may be bone marrow-derived progenitor and stem cells. This paucity of circulating rare cells, in particular in the peripheral circulation, has led many to believe that bone marrow as well as other organ-related cell egress into the circulation is a response to pathological conditions. Little is known about this, though an increasing body of literature can be found suggesting commonness of certain rare cell types in the peripheral blood under physiological conditions. Thus, the isolation and detection of circulating rare cells appears to be merely a technological problem. Knowledge about rare cell types that may circulate the blood stream will help to advance the field of cell-based liquid biopsy by supporting inter-platform comparability, making use of biological correct cutoffs and “mining” new biomarkers and combinations thereof in clinical diagnosis and therapy. Therefore, this review intends to lay ground for a comprehensive analysis of the peripheral blood rare cell population given the necessity to target a broader range of cell types for improved biomarker performance in cell-based liquid biopsy.

## 1. Introduction

A new window has opened into the diagnostic and theranostic exploitation of blood circulating rare cells (CRC) [1,2]. Greater awareness about CRC has been raised by investigations related to cancer cell-based liquid biopsy (cbLB) [1]. Thereby, epithelial markers, mostly CD326 and cytokeratin, were targeted initially in enrichment and analysis, presuming that circulating tumor cells (CTC) are of epithelial nature in carcinomas, melanomas and sarcomas [1]. Nevertheless, the usefulness of cbLB in other diseases has been demonstrated, amongst them cardio-vascular diseases [3], fibrosis [4], inflammation-related diseases [5], diabetes [6], hematologic disorders [7] as well as in reproductive health [8,9]. Therefore, the potentials of cbLB extend beyond their current boundaries. Consequently, in view of accidental findings and occasional investigations on CRC other than cancer-associated cell types, the CRC concept needs further adaption and comprehension [2,10,11].

CRC may be coarsely defined as nucleated cellular events not smaller in size than a few micrometers at concentration levels not exceeding a few thousand events per mL blood. However, there is no official definition about cell rarity with regard to number and type. CRC may only share the aspect of rarity and the complementary association with certain diseases. The composition of different CRC types particularly in peripheral blood may then be referred to as peripheral blood rare cell population (CRCP). The CRCP may comprise “constructive” and “destructive” cell types, respectively. Constructive cell types are common constituents of the CRCP and most likely dedicated to repair or maintain homeostasis thereby being mostly bone marrow-derived, such as multi- and pluripotent stem and progenitor cells [6,12]. Destructive cells are less common or uncommon and allegedly originate from micro-lesions or are product of homeostatic mechanisms for maintenance mostly being mature intact or defective somatic cell types. It has been asserted that health and disease are directly associated with the physiology of rare cell populations foremost in bone marrow and tissue and thus, most certainly in the blood [13]. In view of the hypothesis that any cell type eventually finds its way into the blood stream deliberately or accidentally, CRC reflect to some degree the physiological situation of the tissue of origin. Consequently, changes in the CRCP constitutes the inherent diagnostic wealth that has yet to be explored.

With respect and in realization of the idea of a CRCP, this review intends to comprehend for the first time the possible composition of CRCP in the peripheral blood under physiologic conditions. Also, we intend to identify the “true” biological concentration range in healthy individuals and thirdly, to assess clinical usefulness of each CRC type or in combination with respect to cbLB. This knowledge would largely facilitate the interpretation of cell function for research as well as biomarker translation for early stage diseases. In awareness of more CRC types then herein identified, discussed are the most frequent or commonly reported cell types including megakaryocytes, endothelial cells, epithelial cells, the larger group of fibroblast-like cells, erythroblasts, very small embryonic stem cells, and a few more, amongst them the hematopoietic stem cell (HSC).

As to limit this review to the purpose of cbLB, no other cell sources than untreated blood were investigated in detail then, excluding drug-induced mobilization into peripheral blood, bone marrow and cord blood. This review shall support our efforts and those of others in CRC biomarker translation, potentially opening up new diagnostic and therapeutic possibilities in cbLB.

## 2. Megakaryocytes

### 2.1. Megakaryocyte General Background

The so-called bipotent megakaryocyte-erythrocyte progenitor (MEP) [14] may be described well as the earliest megakaryocyte (MKC) progenitor and arises from differentiation after HSC proliferate and commit to the myeloid lineage [15]. Megakaryopoiesis mainly takes place in the bone marrow with the main function of thrombopoiesis. However, under pathological conditions, extra-medullary megakaryopoiesis is not uncommon [16,17]. It is said that one MKC sheds 2000 to 11,000 platelets per day. Recently, MKC function has been ascribed to immunity within the circulation based on findings of immune-related interactions with other blood cells [18], but also with bacteria [19]. Furthermore, Maroni demonstrated that MKC are essential for skeletal homeostasis, due to the expression and production of the bone-related proteins [20,21,22,23,24]. Kaufman et al. [25] put forward the theory of intra-pulmonary thrombocytogenesis which remains under debate till this day in particular with regard to the proportion of platelets produced in the lung [18,21,22]. Unlike small and indistinguishable bone marrow-dwelling megakaryoblasts, mature MKC are morphologically distinct given their size ranging from 30 µm up to 150 µm in diameter and 50 µm in diameter on average. Also, characteristic are the highly lobulated nuclei. The rather unique cell in particular nucleus morphology was explained by their thrombopoietic function complying with the high synthetic demand of platelets. High platelet generation is is achieved by polyploidy as a result of endomitosis thereby, acquiring a genomic content averaging 16N (range 4–128N) [23,24]. Furthermore, mature cells may be discerned between platelet carrying cells with cytoplasm and those having released their content appearing as cytoplasm-free or naked MKC that may still measure 20 µm in size [11,25,26]. Our investigations on CRC relying on negative selection [27] suggest findings of naked MKC in healthy donor peripheral blood, when compared with published imagery and descriptions of morphology (Figure 1).

Traditionally, a set of markers distinguish differentiation and maturation. Early MKC progenitors exclusively express CD34 and may express CXCR4 marker for being associated with chemotactic signaling that mediates bone marrow migration by CXCL12 [28]. The later differentiation step from the MEP toward the MKC lineage is mainly driven by thrombopoietin and can be traced back by the loss of CD45 expression and the appearance of the fingerprint marker of the MKC lineage, the CD41a marker. The CD41 marker has been reported MKC specific yet, is also expressed in myeloid progenitors. Therefore, hematopoietic CD34/CD45 positive cells gradually express CD41a, CD61 as MKC differentiation progresses. Furthermore, CD61 positive/CD41 negative megakaryocytes were defined as terminally differentiated megakaryocytes that can produce platelets [29].

### 2.2. Circulating Megakaryocytes

Despite a common belief that bone marrow impairments cause MKC egress into the circulation, there remains little doubt about the commonness of circulating MKC hence, being part of the blood circulating CRCP under physiological conditions [26,29,30]. However, sufficient proof of peripheral blood commonness may only exist for the naked MKC. Megakaryoblasts and cytoplasmic MKC were only isolated in the setting of afflicted individuals [31,32]. Nevertheless, Cunin and Nigravic [18] asserted the notion of direct egress from bone marrow sinusoids into the circulation. Dejima et al. [29] investigated the presence of circulating MKC in lung cancer patients finding mature and immature MKC in the pulmonary arterial blood in numbers independent of tumor size, stage and histology, and consequently suggested the strong likelihood of a physiological process.

Accepting a physiologically-driven process, MKC immune function and more importantly, intra-pulmonary thrombocytogenesis is the best explanation for the existence of circulating MKC [21,25,26,29]. It was estimated that 250,000 MKC reach the lung every hour. The common notion is that bone marrow-derived MKC are filtered out by the lung thus, become relative short-lived pulmonary-derived CRC [26,29]. In accordance, the majority of MKC were found in pulmonary artery blood [21,29]. Furthermore, tests on the simultaneous determination of cardiac output in individual subjects allowed a perspective on MKC traffic going to and leaving the lungs [21]. In consequence, a certain fraction of cytoplasm-free MKC will travel within the entire circulation and be measurable in the peripheral blood as naked MKC.

### 2.3. Megakaryocyte Isolation

In general, thrombopoiesis is of clinical interest. Therefore, MKC were mostly isolated by mobilization of peripheral blood progenitor cells or from bone marrow aspirates [33]. Isolation included the selection of low-density mononuclear cell (MNC) fractions that were obtained by density gradient centrifugation methods. Often, the low-density fraction was then subjected to positive selection by magnetic bead technology targeting CD34, followed by cell culture and purification. Typically, FACS was used to isolate cells positive for CD41a, CD42b, CD6l and negative for CD45, CD33, CDl lb and CD15.

With relevance to cbLB and for convenience and practicality, megakaryoblasts and MKC may be best accessed via venous peripheral blood. Herein, naked MKC and megakaryoblasts are most likely identifiable. Megakaryoblasts are assumed to be excluded from the said lung filtration function thus, remaining intact in the circulation and represent a potential cell target. For specific identification of a particular large MKC, one may follow proven and simple blood filtration methods as reported by [26,29,30]. Xu-Lei et al. [11] demonstrated usefulness of a microfluidic chip (Parsortix system) for the isolation of MKC. The chip does not require any pre-enrichment of blood cells. The selectivity of the chip is based on deformability and size of cells. For untouched isolation of a greater variety of cells with respect to physical properties, specialized rare cell isolation technology could be more suited. Such technology has been presented in form of the CTC-iCHIP [2]. Three chip modules carry out cell separation procedures that include red blood cell removal, inertial focusing of the microfluidic cell stream in preparation of the last step, the magnetic separation. The entire make up is based on the negative selection principle, thereby removing undesired cells such as RBC, platelets and WBC consequently, exposing CD45 negative CRC. The authors reported the detection of DRAQ5+/CD41+/concanavalinA−/glycophorin A−/CD45−/CD16− cells and suggested those to be circulating MKC. Similar in principle and suited for MKC isolation are other magnetic cell separation platforms relying on negative selection [27,34]. Moreover, circulating cytoplasmic MKC are best isolated from the right atrium or inferior vena cava using a pressure fixation method [21].

### 2.4. MKC Clinical Usefulness

MKC is a common biomarker in diagnosis of various disorders comprising myelodysplastic syndromes [35], inherited platelet and megakaryocyte disorders [36], myeloid neoplasms and acute leukemia [37]. Mostly, bone marrow biopsy is required for diagnosis, yet also taking into account the peripheral blood blast count that involves megakaryoblasts [37].

It is well to say that the peripheral blood circulating MKC is a neglected biomarker in cellular blood analysis, though it may turn out to hold tremendous diagnostic value [32,38]. Knowing about the commonness, it is not the detection, but rather the elevation or the reduction of circulating MKC concentrations, respectively, that indicate a non-physiological status.

Ever since their discovery in liquid biopsy, circulating MKC elevation has been associated with solid tissue cancers [39]. MKC elevation can be explained by increased megakaryopoisis in the bone marrow or spleen, and increased megakaryopoiesis in turn was associated with bone metastasis [17,20]. To this day, it is unknown whether the MKC is friend or foe in solid tissue cancers and thus, difficult to translate as cancer biomarker [20]. Jackson et al. [17] associated increased megakaryopoiesis (in the bone marrow) with less aggressive metastatic growth suggesting a cancer-inhibitive role of MKC. Actual investigations dedicated to cancer biomarker translation of circulating MKC is sparse. Leversha et al. [40] reported polyploid large cells in the circulation of castration resistant metastatic prostate cancer (mCRPC) patients, suggesting firstly that mCRPC involves bone marrow disturbances and secondly, that circulating MKC are relatively easy to detect in mCRPC patients but not in healthy donors. Xu-Lei et al. [11] found that elevation in circulating MKC levels in mCRPC patients tended to be associated with better survival, thus supporting the findings of Jackson [17]. Interestingly, the authors stated specifically the finding of naked MKC, and suggested a clear difference in prognosis between cytoplasmic and naked MKC Bhakdi et al. [34] included the presence of “giant” polyploid CD45 negative cells into an extended CRC panel, that in combination was used as criterion to predict metastatic prostate cancer. Dejima et al. [29] investigated circulating MKC in lung cancer patients after accidental identification of megakaryocyte-like cells in cancer patients’ pulmonary arterial blood using a filtration method purposed to isolate CTC. The authors concluded that the MKC count (on average 442 per mL) in the pulmonary arterial blood did not correlate with disease stage suggesting less biomarker potential, particularly of pulmonary artery- derived cytoplasmic MKC.

Given the most recent literature, the circulating MKC seems to have been rediscovered for cancer-associated cbLB, yet often by accident [11,29] and sometimes failing to investigate their actual nature [34,41]. Moreover, none of those relevant reports cited herein were able to reproduce MKC commonness in peripheral blood, when compared to past findings [26,30] and may be perceived as a step backwards in MKC biomarker translation. Nevertheless, the association of MKC elevation with in particular metastatic carcinomas may seem reproducible and suggests validity in predicting metastasis and tumor aggressiveness.

Circulating MKC may hold predictive validity in diseases other than cancer. According to past findings, circulating MKC elevation may be associated with disorders of the bone marrow [38] or is associated with sepsis [19] or any non-hematological disorders that may include viral or parasite infections and even burns [32].

At it may seem, the diagnostic potential of circulating MKC addresses many different diseases and exceeds current scope of investigations that is mainly limited to investigations on metastatic cancer. Potentials of advancement may lay in more sensitive platforms and deeper characterization of MKC. Respective knowledge can be derived from investigations of bone marrow dwelling MKC. Winkelmann et al. [42] suggested that MKC ploidy was significantly higher in patients with metastatic disease when compared to control groups however, using post mortum bone marrow aspirates. Furthermore, thrombospondin-1 (TSP-1) MKC status might be an interesting marker as to predict tumor progression [43]. Some evidence hints at a role of TSP-1 in suppressing tumour angiogenesis in the earliest stages of tumour growth. In leukemia or pre-leukemia, essential thrombocythemia (ET) was shown to elicit MKC with a proliferative profile, whereas MKC in myelofibrosis exhibited greater pro-apoptotic impairments [44]. When looking at a single cellular marker, lysyl oxidase may be an interesting biomarker playing role in fibrosis and hinting at progression of myelofibrosis [45]. Similarly relevant to cbLB is the find of elevation of caspase-8/9, Diablo, Survivin and p53 in MKC associated with myelofibrosis [44]. MKC deprivation was associated with immune thrombocytopenia [46] and may likewise be associated with deprivation of circulating MKC.

General interest in MKC for therapeutic applications may lay in thrombopoiesis as to replace platelet transfusion therapy [47]. Hereby, a prospective application is the re-infusion of ex vivo/in-vitro generated autologous MKC as to prevent or reduce chemotherapy- induced thrombocytopenia or treatments in any condition of reduced platelet production [46,48,49]. So far, the in-vivo generation consisted of derivations from HSC [50] and fibroblasts [48] thus requiring lengthy, complicated and low efficiency protocols. The idea of fishing out matured MKC and MK progenitors from blood is certainly not new, yet has not been achieved for the possible reason of its rarity. In our opinion, the existence of MK progenitors in the circulation is most likely and worthwhile to explore for the field of platelet transfusion therapy. A summary of biomarker potentials of MKC has been given in Table 1.

## 3. Endothelial Cells

### 3.1. Endothelial Cell General Background

All endothelial cells (EnC) may be derived from the hemangioblast and shall be discussed as mature and progenitor cells [51]. The function of mature EnC is mostly associated with angiogenesis. Moreover, EnC are able to produce anticoagulants responsible for blood fluidity. Specialization of mature EnC addresses different angiogenic functions and involves three morphological distinct cell types. In need of new blood vessel formation, EnC become activated which are called migratory (i) tip cells that guide the growing vascular sprout and (ii) stalk cells that proliferate and elongate the sprout. Non-proliferative quiescent (iii) phalanx cells regulate vascular homeostasis and provide the endothelial barrier upon maturation of the vessel [52]. EnC may also be classified according to location into venous and arterial, lymphatic or hepatic sinosoidal endothelial cells and tumor-associated endothelial cells. In contrast to EnC, endothelial progenitor cells (EPC) are bone marrow-derived and commonly circulate the peripheral blood in steady-state thereby, contributing to neo-vascularisation [53,54] in tissue repair and maintenance. The attraction of EPC to sites of hypoxic tissue ischemia or tumor growth is a consistent finding that stimulate EPC proliferation and angiogenesis [55]. EPC subtypes may include angioblasts, primordial endothelial cells, hemogenic endothelial cells and vascular endothelium stem cells. Last but not least, EPC have been discerned in early and late outgrowth EPC, which are distinguishable on the basis of their proliferative capacity, measuring 20-fold vs. 1000-fold expansion, respectively. The latter progenitor cell subset is referred to as endothelial colony forming cell. Of note is the insight about possible endo-mesenchymal transition whereby endothelial cells acquire a mesenchymal and stem cell-like phenotype [56].

Several markers have been reported that only in combination may specify the endothelial nature that however, is also common to cells of the monocytic lineage [57,58]. EnC are generally associated with expression of antigens including CD146, CD144, vWF, VEGFR-1/2. Additional markers may include CD31, CD141, c-Kit (stem cell antigen) and CD105. EnC do not express hematopoietic markers such as AC133, CD45 and CD14 and progenitor cell markers, such as CD34 and CD133. However, circulating EnC were associated with varying CD34 expression status [59,60,61], suggesting the need of a more stringent characterization strategy. CD146 was referred to as endothelial cell marker being a member of the immunoglobulin superfamily and was originally identified as a melanoma marker [62]. CD146 may also be found on trophoblasts, mesenchymal stem cells, periodontal tissue, and some malignant tissues. Different activation statuses of EnC may be defined by the presence of adhesion molecules such as CD54 (ICAM-1), CD62E (E-selectin), CD62P (P-Selectin), VCAM-1, CD106 and pro-coagulant (CD142) markers. Activated EnC were also identified by the presence of inflammatory-associated markers; the vascular-adhesion protein-1 (VAP-1) and MHC class I-related chain A [5]. Subtyping EnC may be done by VEGF characterization. Hereby, phalanx cells are found to express high levels of VEGFR1 [63] whereas other types expressed VEGFR-2 (also known as KDR). Tip cell specific markers may include CXCR4 as well as receptors for axon guidance cues, such as the Netrin receptor UNC5B and neuropilin-1. The Ang-2 receptor, Tie-2 was also found to be expressed by stalk cells, while not detectable in tip cells [52]. Of note, tumor-associated circulating EnC were reported to overexpress CD276, an immune checkpoint molecule therefore being distinctive to normal circulating EnCs [64].

Previously, EPC were distinguished from EnC based on CD133 expression, for example using the phenotype CD45−, CD146+, CD133+ for circulating EPC as in contrast to EnC using CD45low or negative, CD146+, CD133- phenotypes in particular for tip EnC. Moreover, EPC were shown to express CD31, vascular endothelial cadherin, von Willebrand factor, and VEGFR-2 [57,58]. There is however, no consensus on reported expression profiles thus, hampering clinical evaluation of EnC and EPC [59]. Late EPC may be characterized in particular by the expression of CD31, CD34, NP-1, c-Kit and KDR and in contrast to early EPC by the explicit absence of CD45 expression. Vascular EnC may be defined as lin−, CD31+, CD105+, Sca1+, CD117+, c-Kit+ phenotype [65]. CD31 may play an important role in staging EPC suggesting later stages when expressed and typically occurring in the blood circulation when compared to bone marrow EPC. Primitive angioblasts were characterized by expression of CD45, CD34, c-kit, lin, VEGFR2 and the absence of AC133 [66]. However, hematopoietic EPC do not give rise to endothelial progeny [67].

In contrast to EPC, EnC are relatively easy to spot, even under bright field vision. For their identification standard H&E staining suffices given their unique morphology and large size [68]. The cells may measure in diameter up to 30 µm. For the trained eye, their subtypes have distinct morphologies. Figure 2 illustrates cells from our data set of rare cells with different morphologies suggesting endothelial cell character. Hereby, a typically high-density large spindle- shaped nucleus in size of >9 µm can be observed with very low yet varying N/C ratios ascribed to the highly variable cytoplasm in size and shape.

### 3.2. Circulating Endothelial Cells

The existence of EnC and EPC in the circulation is undisputed given the abundance of mentions in supporting literature since the 1970s or perhaps earlier [69]. Nevertheless, a great difference with respect to origin distinguishes EnC from EPC. Circulating EnC may not be blood-native rather being a consequence of intimal injury, leading to detachment of EnC from the basement membrane. In contrast, circulating EPC may have entered the blood stream naturally via a similar process when compared to HSC egress from the bone marrow [70]. Evidence of mobilization into the blood stream upon certain stimuli including tissue ischemia has been given [6]. Therefore, circulating EPC are blood native and common due to a deliberate physiological process of egress [66].

### 3.3. Endothelial Cell Isolation

Circulating EnC and EPC have been adopted as potential targets for cbLB. The standard method to isolate endothelial cells in particular colony forming endothelial cells relies on the separation from MNC by cultivation [59,71]. EPC may be further pre-enriched prior to cultivation by magnetic cell separation based on positive selection of CD146+ [72] or CD133+ cells. In this context, microfluidic technology was also proposed with the intention to isolate endothelial colony-forming cells by virtue of CD34 expression [67]. EPC may be best cultivated by plating enriched CD34+ cells or pre-enriched MNC into collagen-coated plates containing EGM-2 medium [73]. EPC and EnC analysis by flow cytometry or microscopy may require purification when derived from blood samples directly that would exceed pre-enrichment efforts. In this sense, Ali et al. [60] employed the CellSearch assay that was modified to analyze circulating EnC. The assay relied on positive selection of CD146+ cells followed by fluorescence microscopy. Hansmann et al. [74] presented the “EPC capture chip” relying on positive selection of CD34+ cells and required only 200 µL blood for sample analysis. Due to variable marker expression in EPC and EnC, in particular with respect to CD34, positive selection introduces a bias inevitably leading to cell losses. A less biased enrichment approach would be more suited. Bhakdi et al. [34] employed a negative selection strategy thereby depleting WBC and residual red blood cells by indirect magnetic labeling and high gradient magnetic separation subsequent to RBC lysis which was followed by a multiplexed cryo-immunostaining method relying on endothelial markers CD31 and CD34. Furthermore, exhaustive enrichment may not always be necessary. Circulating EnC were simply identified, characterized, and quantified by flow cytometry using a stain-lyse-no-wash single-platform method [75]. Bethel et al. [68] reported the isolation of endothelial cells without prior enrichment solely relying on RBC lysis and specialized fluorescence microscopy deemed the HD-CEC method.

### 3.4. Endothelial Cell Clinical Use

In general, endothelial cells are investigated on grounds of their functionality and relate most of all to vascular diseases. In general, the level of circulating EnC is persistently higher upon pathological when compared to physiological conditions [34,68,76]. Consequently, circulating EnC have been associated with vascular injury, while circulating EPC have been associated with re-vascularization and endothelial regeneration (D) [77]. Circulating EPC were more frequently investigated in peripheral blood when compared to circulating EnC which can be ascribed to differences in concentration levels (chapter 9). Dysregulation of pathways to control angiogenesis is frequently observed in the disease setting and warrants potential diagnostic biomarker applications [5,59].

Despite the great non-specificity of EnC towards to pathological conditions, their validity as biomarker has been investigated exhaustively in cancer cbLB for various applications [34,54,60,61,78]. It is believed that either injury-related inflammation at the tumor site or tumor neo-vascularization causes EnC elevation [79]. Bhakdi et al. 2 [34] investigated the biomarker usefulness of a combination of epithelial CTC and tumor-associated endothelial cells based on an in-house test to predict malignancy in male individuals with clinical suspicion of prostate cancer. The authors reported a doubling in positive predicted value when used as add-on to common PSA testing. Kraan et al. [61] and Ali et al. [60] investigated the usefulness of circulating EnC for the prediction of treatment response to pre-operative chemotherapy in patients with operable breast cancer. The high relevance can’t be more stressed for the purpose of monitoring tumor evolution during and after therapy, potentially leading to improved adjuvant therapy decision making. The authors noted the significance of the ratio between CD34+ and CD34- EnC wherein a higher degree of CD34+ EnC was associated with non-pathological response suggesting different underlying pathologies for the two cell subtypes [60]. Furthermore, CD146 expression was reported to promote cancer progression by induction of epithelial–mesenchymal transition and thus, may be useful for cancer prognostication [80,81,82]. This notion was supported by Ilie et al. [78] who investigated the usefulness of CD164+ circulating EnC counting in combination with CD164 serum level detection for predicting clinical outcomes in non-small cell lung carcinoma patients undergoing surgery, showing that marker elevation in both cases at baseline correlated positively with poor prognosis. Similar investigations with respect to the detection platform and outcome have been conducted for breast and colorectal cancer [60,83]. Moreover, Rahbari et al. [83] compared circulating EnC with epithelial CTC for prognostic performance in metastatic colorectal cancer using the CellSearch system. The authors reported a higher hazard ratio associated with overall survival (OS) in case of circulating EnC when compared to CTC. Nevertheless, the overall benefit of circulating CTC and EnC in this liquid biopsy application stands in question as the hazard ratio values are low and outperformed by the serum marker CA 19-9 [83]. Of note is one report that seems to disagree with the general notion of worse prognosis upon circulating EnC elevation at baseline using the same technology when referred to Ali et al. [60,84]. The authors reported a positive correlation of baseline counts with good prognosis. After all, in congruence with other works, a decrease in both circulating EnC and EPC correlated with a longer PFS or OS in the majority of studies, before and after effective anti-angiogenic therapy and/or chemotherapy [85,86] Polyploidy is a common cellular cancer marker and has been included into investigations on circulating EnC [34,87,88]. Latest work on the topic may come from Zhang et al. [88], employing Cytelligen’s SE-iFISH to identify PD-L1 positive, aneuploid circulating EnC (multiploidy in chromosme 8) assumed to be tumor-associated and investigating their usefulness as companion diagnostic as to predict resistance to checkpoint blockade immunotherapy in advanced NSCLC patient [88]. The authors reported that patients with multiploid PD-L1+ tumor-associated circulating EnC were found to have a statistically significant shorter PFS when compared to the subjects without. The study is preliminary for influencing treatment decision making in late stage NSCLC patients on grounds of advanced cbLB technology. Of note is that the same group reported in an earlier study [87] that aneuploidy may not be specific to malignancy at all having detected polyploid endothelial cells in a majority of healthy donor endothelial cells. Polyploidy in circulating EnC may be physiological after all and related to senescence [89]. Thus, clinical interpretation of endothelial polyploidy entails careful biomarker definition and establishment of robust specificity cutoffs. Besides, polyploid endothelial cell-targeted LB may be useful for investigations on human aging [90,91]. There are also studies dismissing biomarker usability of certain circulating EnC. Gootjes et al. [64] investigated the CD267 positive circulating EnC (basic phenotype: CD34+/CD45neg/CD146+/DNA+) to predict therapy responsiveness to palliative systemic therapy in patients with metastatic colorectal cancer and reported inadequate prediction potential. Consequently, prognosis seems to remain the most valid biomarker application upon baseline measurement.

Fewer studies investigated the correlation of circulating EPC with cancer. Rhone et al. [92] investigated the association of circulating EPC (CD45−, CD34+, CD133+, CD31+ with common cancer determinants in non-metastatic breast cancer concluding that elevation was positively correlated first of all with cancer disease, when compared to a healthy cohort, with patient age over 60, with positive Her-2 status, and was negatively correlated with histopathological grading. The correlation whether positive or negative of circulating EPC with cancer seems to be under debate. The authors speculated that this marker would be useful in rating aggressive neo-angiogenesis. Circulating EPC elevation with phenotype CD144+/CD34+/CD133+/CD45− was found in hepatic carcinoma patients and seemed to be positively correlate with tumor burden, suggesting potentials in cbLB biomarker translation (see HSC) [93].

Apart from solid tissue cancer, circulating EnC and EPC have been associated with many other pathologies and this constitutes a serious pitfall when used as independent markers in cancer diagnosis. In the know of an incomplete list, endothelial elevation may occur upon ischemia, congestive heart failure, myocardial infarction (MI), sepsis, vascular trauma, hypertension, hyperlipidaemia, and hyperglycaemia, acute and chronic infection [94,95], sickle cell anemia, vasculitis, diabetes type II [96] deep vein thrombosis) [75,76], granulomatosis [5], pulmonary hypertension [74] or cirrhosis [97]. In inflammation, circulating EnC and EPC may interact with each other where inflammation activated circulating EnC induce EPC dysfuntion [5]. Consequently, it is worthwhile to investigate several phenotypes at the same time to obtain greater picture of the disorder [96]. In contrast to mature EnC, EPC numbers were often reported to be inversely correlated with disease [98]. One of the big-interest marker applications beside cancer is the prediction of MI related death as well as cardiovascular risk assessment [99]. Hill et al. [99] reported an inverse correlation with disease severity which was reproduced by Werner et al. [100]. Beside circulating EPC deprivation, the authors reported impaired cell functionality which was suggested to be an independent predictor of death. In line with Werner [100], Schmidt-Lucke et al. [101] reported circulating EPC deprivation in coronary artery disease and concluded that EPC quantification would be an independent predictor of atherosclerotic disease progression. EPC deprivation in numbers could mean a reduction by half which was found when comparing pulmonary arterial hypertension patients with healthy controls [74]. In diabetes, circulating EPC were reported to be decreased [59], but also increased [96] therefore requiring a closer look into the patient status. Circulating EPC elevation may indicate diabetes type 2 related peripheral neuropathy [96]. A similar controversy seems to persist about liver cirrhosis that was reported to elicit EPC deprivation by Chen et al. [102], yet more recently, was associated with EPC elevation [97].

Interestingly, circulating mature EnC have been rarely mentioned or investigated for non-cancer diseases. Sabulski et al. [103] recorded circulating EnC daily dynamics purposed to investigate the usefulness of predicting complications in pediatric patients undergoing hematopoietic stem cell transplant. The small sample size study showed elevation in patients within 2 weeks after treatment when compared to negative controls. Despite the authors suggestions, peaking did not seem to discriminate severity in complications including death.

In transplantation medicine, circulating EPC are of interest for their inherent vasculogenic properties ideally suited for autologous vascular or so called pro-angiogenic therapies [104,105]. One may envision the restoration of blood flow to ischemic limbs in diabetes or in MI [66,106]. However, the field is hampered by a lack in standardization of phenotypes and effective cultivation and isolation methods [59].

Last but not least, objective lifestyle biomarkers may become increasingly important in our health conscious modern societies. Physiological age as well as smoking was associated with a reduced amount and function of circulating EPC [90,107]. Recently, Magalhães et al. [108] employed circulating EPC and EnC quantification to evaluate exercise-induced vascular adaptations and a method to attenuate those and concluded that vascular damage was inflicted by the physical practice.

In conclusion, the mature EnC has been proven useful in cancer cbLB with regard to quantification. In contrast, the circulating EPC has been correlated inversely to a series of non-malignant vascular diseases. However, the diagnostic wealth may lay in more differentiated cell clustering as well as in investigating cell functionality. In particular circulating EnC may reveal more specific information about certain types of injury by the presence of either one or all of the three subtypes phalanx, tip, and stalk- cells, respectively. Furthermore, the activation status of circulating EnC is awaiting investigation and may fit into certain clinical settings, in particular in association with inflammation. Clinical implications that are associated with the various subtypes remain vastly speculative.

## 4. Erythroblasts

### 4.1. Erythroblast General Background

Erythroblasts (EB) are classified as bone marrow-native and are progenitors of the red blood cells (RBC). Approximately 2 × 10^11^ new RBC are generated per day in a process called erythropoeisis taking place in the red bone marrow within erythroblastic islands. This RBC maturation process starts with the erythroid committed pro-erythroblast and ends in the blood stream where reticulocytes complete maturation within 1 or 2 days. The maturation process is typically classified according to morphological changes that mainly relate to cell size reduction and chromatin condensation giving rise to basophilic (Baso), polychromatophilic (Poly) and orthochromatophilic (Ortho) EB [109]. Finally, enucleation and the loss of all organelles follows giving rise to immature reticulocytes. In the periperhal blood, EB are often referred to as nucleated red blood cells (NRBC).

Morphology plays an important role in the qualification of EB maturation stage. EB are distinct yet typically blast-like cells often round, measuring 7 up to 25 µm in diameter in late and early stage, respectively. The nuclei are also round and measure in diameter between 4 µm to 9 µm also depending on maturation stage. Due to the correlation between morphology and maturation, the nucleus to cytoplasm (N/C) ratio is an important parameter to classify the maturation status of an EB. Usually, early maturation stages as represented by the pro-erythroblast yields a low N/C ratio that increases to near 1 for Ortho-Ebs also often referred to as normoblasts. Figure 3 illustrates circulating EB at different maturation stages. In general, erythroid precursors can be characterized by immuno-phenotyping using a set of cell surface markers that include the transferrin receptor antigen CD71, GPA, Kell blood group protein, integrin associated protein CD47, and the glycoprotein antigen CD44. High expression of CD71 and GPA are unique to cells related to the erythroid lineage [110]. Of note is that primitive EB may not or weakly express CD71 [111]. Moreover, the CD147 is a plasma membrane protein with a function as extracellular matrix metalloproteinase inducer and was reported to be expressed amongst many other cell types on immature EBs [112].

### 4.2. Circulating Erythroblasts

The appearance of NRBC in peripheral blood has often been postulated to herald severe disease including poor prognosis [113]. Nevertheless, NRBC are commonly present in the peripheral blood even in healthy individuals. Prove has been provided consistently throughout the decades by non-invasive pre-natal testing (NIPT) [114,115] and recent investigations on rare cells in peripheral blood [2,10] Most explanations for the presence of NRBC in the peripheral blood relate to impaired erythropoiesis that has been divided into three categories; inefficient erythropoiesis for example caused by thalassemia, stress induced erythropoiesis for example erythropoietic resumption after chemotherapy, and pathological erythropoiesis due to primary abnormalities in hematopoiesis (for example caused by various leukemias). In general pathological erythropoietic activity is considered a consequence of hypoxic stress [113,116]. The presence of such cells under physiological conditions still remains to be elaborated that may be accidental or in fact assume steady-state blood circulation with unclear function and unknown maturation fate. Given the abundance in findings even though at low concentrations, a regular or physiological traffic between circulation and bone marrow can be expected. Unusual may be the finding of NRBC at different maturation stages suggesting a non-accidental [10]. In fact, immune activity has been postulated as well as the ability of EB to actively leave the bone marrow [117]. In conclusion, the finding shall indicate that Ebs are in fact normal in the blood and a common part of the CRCP.

### 4.3. Erythroblast Isolation

The gold standard EB identification method is direct evaluation through light microscopy of May-Grünwald-Giemsa stained blood smears in pathology laboratories. However, most commonly automated analysis via hematological analyzers is performed as to overcome drawbacks such as insensitivity and intra- and inter reader variation. Apart from differential cell count, different isolation methods were proposed in particular for fetal cell analysis in the field of cell-based NIPT. Herein, NRBC were subjected to positive selection using a-GPA magnetic cell separation technology subsequent to pre-enrichment by ficoll density gradient centrifugation [118]. Alternatively, NRBC were enriched by biotinylation of erythropoietin and subsequent selection by strepavidin magnetic particles [8]. A high purity microfluidic platform was introduced by relying on a two-step enrichment process that was based RBC hyper-aggregation thereby fractionating erythroid cells and a subsequent negative enrichment process as to remove all leukocytes from the erythroid cell fraction. Wei et al. [9] presented an in-house herringbone microchip relying on CD147 capture of EB subsequent to pre-enrichment of MNC. Captured EB were obtained by enzymatic release. Circulating EB might as well be isolated by density gradient centrifugation alone. Kwon et al. [119] compared density gradient modalities and concluded that high EB yield can be achieved by optimized osmolarity and double-density gradient system. Negative selection magnetic particle technology was proposed for efficient EB detection [2,10,120]. More recently, Schreier et al. [10]. relied on RBC lysis followed by macroscale automated rare cell enrichment and [2] employed the CTC-iCHIP platform, a microchip that firstly removes RBC and then separates magnetic from non-magnetic material. Apart from enrichment and isolation, NRBC may be simply cultured. Chen et al. [115] cultured erythrblasts after Ficoll cell plating in a culture medium containing a low concentration of erythropoietin. Colonies emerged after 10 days.

### 4.4. Erythroblast Clinical Use

Damage or stress to bone marrow is commonly accepted to induce NRBC elevation in peripheral blood thus, being useful in general as biomarker for bone marrow *abnormality (9)* [121]. Consequently, NRBC elevation can be found in many pathological conditions and most of all cancers involving the bone marrow but also in critically ill patients. Commonly, NRBC count is part of the complete blood count in clinical hematology. However, the sensitivity of standard hematology analyzers as well as manual differential blood counting is rather low [122]. Threshold concentration levels relate to NRBC abundance rather than rarity and can then be considered already highly abnormal when compared to physiological conditions. Thereby, NRBC detection was strongly associated with chronic myeloid leukemia, acute leukemia and myelodysplastic syndromes [123]. Counts may then rise up to 4.45 × 10^5^ NRBC per mL as for example reported in case of idiopathic myelofibrosis [122]. A more comprehensive list of hematological disorders relating to NRBC elevation was given by Rin et al. [122].

It should be noted that NRBC biomarker development in the lower concentration range remains to be largely explored. Even though, clinical significance of low NRBC concentrations <1% remains under debate [124,125] we argue that counts above the physiological concentration and/or subtle but definitive concentration changes hold clinical information. NRBC clinical usefulness may relate to early disease detection [125]. Herein, disease specific diagnostic testing would be prompted upon NRBC elevation in asymptomatic individuals or patients with inconclusive and non-specific symptoms. The patient benefit of early detection may then lay in improved OS so mentioned in cases of myelofibrosis [124,126]. Also NRBC count may be useful in therapy monitoring, specifically to predict therapy failure. Phan et al. [127] reported usefulness in CML patients treated with imatinib. An association was found between BCR-ABL transcripts in CML bone marrow cells and likewise high BCR-ABL expression and high NRBC counts in the peripheral blood. Remission failure was predicted by high NRBC counts throughout therapy and seemingly good responders had NRBC counts below a given detection threshold. However, the diagnostic accuracy only measured only 0.64 AUC. We hypothize that the AUC value could be significantly increased by more sensitive NRBC detection platforms. Moreover, it is to be noticed that the NRBC biomarker potential remains to be investigated in solid tissue cancer cbLB.

In non-oncological diseases, counting NRBC may find usefulness as independent prognostic biomarker for OS in critically ill patients. Desai et al. [128] reported elevation in NRBC in ICU patients with surgical sepsis being associated with higher mortality and was found useful as early and independent prediction biomarker for OS. Menk et al. [129] investigated the NRBC count in acute respiratory distress syndrome. In line with Desai et al. [128], Menk et al. [129] showed NRBC detection per se to be an independent risk factor for mortality with a doubled risk for ICU death. NRBC value at ICU admission was found to be an independent risk factor for mortality. A cutoff level of 220 NRBC/μL was associated with a more than tripled risk of ICU death. Similarly, Monteiro et al. [130] proposed amongst other hematological laboratory parameters, NRBC count usefulness in monitoring ICU cardiologic patients as to predict in-hospital mortality and showed a clear association with counts in range form greater 0 to over 200/uL. However, the positive predicted value was low reporting 26.8%. We entertain the thought that perhaps finer cutoff intervals could be facilitated by more sensitive detection platforms and might increase marker specificity towards a life threatening situation. An alternative to the one-time count is the assessment of dynamic changes in concentration levels. As a marker of impending demise, NRBC “appeared” on average 9 days before death upon daily screening in patients in surgical intensive care units) [131]. Greater sensitivity towards NRBC detection might extend the given 21 day period of prediction. In conclusion, the NRBC or circulating EB seemed to have tremendous potential in disease characterization, yet are vastly unexplored for use in early stage or mild bone marrow disorders.

## 5. Fibroblast-like Cells

The term fibroblast-like cell commonly denotes a very heterogeneous hematopoietic and/or mesenchymal cell population. The herein discussed fibroblast-like cells shall be classified into mesenchymal stromal/stem cells (MSC), fibroblasts (FB) and fibrocytes (FC). An illustration of seemingly different peripheral blood circulating fibroblast-like cells is given in Figure 4. The mentioning of fibroblast-like cells dates back to Friedenstein’s group that discovered colony forming unit fibroblasts isolated from bone marrow tissue [132]. Ever since, scientists struggle to provide clear definitions of this cell class which consequently, negatively affects advancement of biomarker validation.

### 5.1. Fibrocyte General Background

Fibrocyte identification may date back to Bucala [133] having detected CD34+ and vimentin+ fibroblast-like cells in culture. FC represent a distinct and common constituent of the peripheral blood, are bone marrow-derived and arise from a monocyte precursor. The hematopoietic origin of human FC is reflected by their expression of CD45 and/or Vav1 [134]. The main idea of being derived from monocyte lineage comes from the expression of CD11b, CD13 and CD14, whereas the production of connective tissue matrix components including collagen-1, collagen-III, and vimentin alludes to the mesenchymal lineage. Therefore, FC are often referred to as leukocytes that mediate tissue repair by producing extracellular matrix (ECM) components as well as ECM-modifying enzymes thus, being actively associated with fibrosis [135]. The combination of markers CD34+/CD45+/pro-collagen-1 identifies peripheral blood FC, but also a subpopulation of macrophages consequently arguing in favor of a common progenitor population. FCs may be distinguished from macrophages by the expression of fibroblast specific protein 1 and the absence of the F4/80 marker [136]. FC lack lymphocyte marker expression CD3, CD4, CD8, CD19, and CD25 both in vivo and after in vitro culture. FC share a role in immunologic responses which is reflected by their expression of chemokine receptors such as CCR2 and CXCR4, by their expression of proteins important for host defense (CD16/32, CD163) and antigen presentation (major histocompatibility complex I and II, CD80, and CD86). Moreover, FC posses multi-lineage differentiation potential [137]. Of note is that the circulating part may be phenotypically limited or different when compared to tissue dwelling fibrocytes [138]. FC activation is key to cell type interpretation. The activation status correlates with tissue residency. Activated FCs can be distinguished from their inactive state by phenotype as well as morphology. Activated FCs express in particular higher levels of phospho–STAT-5, STAT-1, JNK, and AKT (G) [139]. Kao et al. [140] observed a morphological change upon activation that is reflected by small oval-shaped to elongated spindle-shaped cells only in the late stage after roughly 5 to 10 days of cultivation. Nevertheless, FC cytoplasm as well as nuclei are commonly described as spindle-shaped [141]. To be more precise, the cytoplasm may appear under microscopy as thin long threads. Once activated, fibrocytes may be indistinguishable from FB [134,142] that is ascribed to the protein synthesis causing swelling of the cytoplasm and constitutes a pitfall in correct identification. Also, FB activation may coincide with the reduction in expression levels of CD34 as well as CD45 [140,143]. This change in expression levels may cause confusion with respect to the existence of a rare FC-like sub-type that was identified in wound tissue of mice being negative for CD45 and CD11b expression, yet being of hematopoietic lineage [134,144]. CD45 negative FC may comprise only a fraction of normal FC. A 5% frequency amongst all FC was reported and would merit the qualification of rarity. Suga et al. [134] made a clear distinction to MSC/FBs by proving the hematopoietic origin as indicated by Vav1 expression. The assertion that FC CD45 expression is maintained throughout maturation/activation may not hold truth following the motto “you get what you select”. Therefore, the question stands to be proven, if CD45 negative FC are in fact activated FC that have lost CD45 expression or a distinct subtype. However, given the fact of commonly found activated CD45+ circulating FCs and the extreme heterogeneity of wound repair cells, we argue that the likelihood of a stable circulating CD45 negative FC subtype is within reason [139,145]. At this point, we would also like to hint to the in Figure 4 shown CD45 negative fibroblast-like cells in particular cells C and D that would correspond to descriptions of non-activated FC. FC can differentiate into tissue myofibroblasts, express alpha-smooth muscle actin (α-SMA), and become less distinguishable from other organ cells, such as stellate cells and fibroblast-derived myofibroblast [133,136,146,147,148]. The general problematic of a great marker overlap between FC and FB is evident. In fact, FC were considered progenitors of FB thereby representing the activated state of otherwise inactive FC [149]. Nevertheless, both cell types, FC and FB are meanwhile considered distinct populations, also in activated cell states. Moore et al. [149] suggested to distinguish both types by the CCR2 protein and upon CCL2 chemataxis. Furthermore, in contrast to cultured dermal FB, cultured FC are associated with a prolonged and markedly upregulated expression of chemokines (MCP-1, MIP-1*α*, and MIP-2*α*), pro-fibrotic factors (TGF-*β*1 and PDGF-A), and angiogenetic factors (VEGF-*α* and b-FGF) [140]. The main cause of confusion is the reduction or loss of CD45 receptor and increased collagen I expression [150,151], so that characterization of FC has been difficult by conventional marker sets. Consequently, confusion lays around the origin of myofibroblasts that can be derived from hematopoietic FC as well as mesenchymal FB. [134,143].

FC are best isolated as described by Quan et al. [135] from buffy coats. Nucleated cells were purified by Ficoll Hypaque density-gradient centrifugation, washed and kept in DMEM supplemented with 10%, heat-inactivated fetal bovine serum. Fibronectin pre-coated culture surfaces were reported to increase yields. After 2 days, the non-adherent cells (largely T cells) are aspirated off, and the remaining adherent cells cultivated for 14 days. Over time, the contaminating monocytes die off, and FC appear as clusters of stellate, elongated or spindle-shaped cells that show long cellular processes. More recently, conventional CD34+ positive selection magnetic cell separation technology was employed for fibrocyte isolation followed by FACS [152]. The authors also proposed a faster fibrocyte quantification method relying on cell surface staining prior to flow cytometry analysis [152] Nevertheless, intra-cellular staining is still needed for in-depth biomarker characterization [138]. In the know of the existence of CD45negative circulating FC subtypes, we propose negative selection by CD45 depletion as improved means of cell purification with subsequent proposed cultivation protocols.

### 5.2. Mesenchymal Stem Cells General Background

There is a seemingly vast heterogeneity of stem and progenitor cells with stemcell-like characteristics that agree with characteristics of the mesenchymal lineage. Such cells may share functionality of tissue repair and replacement [153] More recently, MSC were associated with immuno-modulatory capacity which includes the inhibition of proliferation and function of various immune cells [154]. The International Society for Cellular Therapy (ISCT) has defined MSC as bone marrow-derived cells with ability to adhere to amorphous plastic surfaces in standard in-vitro culture conditions, shall express defined surface antigens and possess multi-lineage differentiation potential that includes adipocytes, osteoblasts, and chondroblasts in-vitro [155]. With ongoing research, the multi-lineage differentiation capacity of MSC was found to include neuronal, lung or hepatic cell types. However, the stringent definition excludes non-adherent MSC-like cells [156]. Also, the definition is diluted by the fact that MSC occur in other tissues apart from bone marrow. MSC-like cells were isolated from muscle, trabecular bone, adipose tissue, placenta, dental pulp, synovial membrane, peripheral blood, periodontal ligament, endometrium, umbilical cord, and umbilical cord blood. In particular, reports about MSC in peripheral blood makes this cell type interesting for cbLB [157,158]. Historically, dermal wound mesenchymal fibroblast-like cells are thought to be derived primarily from local recruitment and proliferation of resident FB. However, greater participation of circulating fibroblast-like cells has been suggested in wound repair. Egress was speculated to be associated with hypoxic conditions [159]. To date, the existence of short-lived bone marrow-derived peripheral blood circulating MSC is undoubted) [160,161,162]. However, MSC egress under physiological conditions requires more investigations. Recently, evidence of MSC in healthy donor peripheral blood was brought forth by Lin et al. [156], who investigated the effect of different culture conditions on the proliferation and differentiation of peripheral blood-derived MSC. The authors argued that adult stem cells patrol the bloodstream and circulate through peripheral organs physiologically and pathologically, which may be required for tissue homeostasis and maintenance. Given the evidence, peripheral blood-derived MSC may circulate in steady-state, thus being common part of the CRCP [163].

Origin dependency of MSC with regard to pheno- and genotype is quite likely. MSC isolated from different types of tissue share some common identification markers, yet their differentiation abilities and gene-expression profiles may vary [164]. Moreover, mesenchymal progenitor cells show similar phenotypic and stem cell-like characteristics, when compared to MSC such as self-renewal multi-potency, the expression of embryonic transcription factors, but elicit different tendencies in proliferation and differentiation when induced. Due to the confusion in cell definition and owing to simplicity in the text, we have addressed all related mesenchymal cell types as mesenchymal stem cells (MSC).

As it may turn out, detailed investigations on the morphology of circulating fibroblast-like cells from peripheral blood are rare. Cultivation may not give a clear answer as to how the cells look like within the circulation having transformed once adhered to the plastic dish. In culture, MSC proliferate and differentiate assuming spindle-shaped morphology and it stands to question if the fibroblast-like morphology is native to the cells and consistent throughout activation status. Kassis [165] isolated mostly fibroblast-like shaped cells from mobilized peripheral blood using fibrin coated particles, suggesting that MSC occur already in this morphology. However, the blood donation was a result of G-CSF-mobilization and therefore may elicit changes in cell activation status. On the other hand, reports exist about large flat, round MSC-like cells in culture [166]. One may argue that fibroblast-like cells may appear initially as round cells and assume fibroblast-like shapes in culture [167]. Large round cells also have been described. More information about the native form of mesenchymal fibroblast-like cells may be best provided by investigations of the bone marrow. Hauser et al. [166] cultured bone marrow stroma aimed to retain morphology of in-vivo conditions and identified fibroblast-like cells as flat large cells up to 100 µm in diameter. Interestingly, a change from spindle-shape to round morphology may be induced by culture conditions as done by supplementation of with dexamethasone (10–7 mol), ascorbic-acid-2-phosphate (0.05 mmol), and β-glycerophosphate (10 mmol).

Phenotypical characterization of MSC has developed into a comparatively creative field possibly due to the lack of definite antigens. Aspects of cell status of in-vivo or in-vitro and furthermore, of the tissue origin were considered. Most agreed antigens of bone marrow-derived MSC in vitro include CD73, CD90, and CD105 while being absent of expression of hematopoietic, epithelial and endothelial antigens CD45, CD14, CD11b, CD54, CD56, CD79a, CD19, CD133, CD324, CD326, CD344, HLA-DR as well as major histocompatibility complex II. Other general markers are the SH antigens 1 to 3 of which, SH2 is CD105 (endoglin). SH1 and 3 may be more specific to MSC. In culture, less common markers included CD108 (SemaL), CD109 (platelet activation factor), CD117 (c-kit), CD166 (ALCAM), CD318 (CDCP1), CD340 (HER-2), CD349 (frizzled-9), SSEA-4, and HLA-CL I. MSC may also express CD10, CD13, CD44, usually fibronectin, collagen (I, III and IV) and adherent cells may express Stro-1, CD49a, CD63a and CD106 (vascular cell adhesion molecule-1). In tissue, MSC have been identified as or may be pericytes aka Rouget cells or mural cells, mesangial cells in the kidney and Ito cells in the liver that closely encircle endothelial cells in capillaries and micro-vessels [168,169]. Crisan et al. [167] showed that pericytes and MSC share the same phenotype and functionality and concluded that the cells are one and the same. Apart from commonly found MSC phenotypes, the expression of pericytes may include CD146, NG2, alpha-SMA, CD140b (PDGF-Rb), and alkaline phasphatase in the absence of hematopoietic, endothelial, and myogenic cell markers. It was only recently that circulating MSC can be derived from non-mobilized peripheral blood at high success rate [157]. Hypoxic culture conditions have been proposed to the do the trick which awaits reproducibility. Phenotypical differences between human peripheral and bone marow-derived MSC may exist with the latter being negative for CXCR4, Nestin, Nanog and Lgr5, a cell surface protein. In contrast, Lgr5 was mainly expressed in peripheral blood derived MSC [157] which suggests usability as stratification marker between the two MSC subtypes.

Kuznetsov et al. [169] self-proclaimed to be the first to discover circulating FB as being genuinely mesenchymal, that differed in the expression of Stro-1, MUC-18, CD105 and alkaline phosphatase when compared with MSC. Stro-1 was discussed as stringently native to MSC, yet under the obligation of plate adherence in culture. Therefore, Stro-1 negative MSC may be indeed characteristic of circulating MSC/FB subtypes. Despite greater similarity, MSC and FB are considered to be two distinct cell types to this day [153,170,171]. However, the exact cells giving rise to FB are unknown [172,173]. The lack of information on the fibroblast lineage has been similarly pointed out by Abercrombie et al. [173] as well as Shamis et al. [174]. While, most of the scientific community sought to identify differences between said cell types, some have looked into their similarities [175,176]. In fact, it has been reasoned that FB and MSC are one and the same cell type in tissue having concluded that there are no differences in culture-derivation methodology, morphology, cell surface marker expression patterns, differentiation potential and gene expression signature that consistently and unequivocally distinguish ex-vivo culture expanded MSC from FB [176,177]. MSC alike, FB have been isolated from skin, adipose tissue, cardiac tissue, cornea, muscle, etc. and share the same functionality of repair and replacement. Yet, despite all similarities in vitro, the in-situ situation with respect to the origin may reveal differences. Engraftment studies in particular for cancer and heart ischemia demonstrated distinction between bone marrow-derived MSC and bone marrow-derived FB [178] In view of the before mentioned, MSC are less differentiated stem cells whose progeny are more differentiated FB [156,179,180].

Commonly, circulating MSC are isolated by ficoll density gradient centrifugation than plated at low densities (1 × 10^6^ per 35 mm^2^ plate) [168]. Zvaifler et al. [180] isolated adherent fibroblast-like cells from buffy coats of normal human blood. The buffy coat was subjected to density gradient centrifugation twice. Then, an elution method was employed to purify them according density using a flow system hereby targeting in particular monocyte-rich fractions. Zvaifler et al. [180] discussed that culture conditions in its very detail may be important to what is actually growing and how cells may look like. Very recently, Lin et al. [156] successfully cultivated MSC from whole blood nucleated cell suspensions that were derived from 10 mL RBC lysed blood samples in a minimum essential medium (a-MEM; Gibco/Invitrogen, Thermo Fisher Scientific, Grand Island, NY; USA) supplemented with 17% foetal bovine serum (FBS; Gibco-Invitrogen, Waltham, MA, USA) and 1% antibacterial agents. The nucleated cells were seeded at a density of 5 × 10^5^ cells/cm^2^ and cultivated in normoxic and hypoxic conditions. With regard to the choice of pre-enrichment methods, a significant lower loss may be expected from using RBC lysis when compared to Ficoll density gradient centrifugation [181,182]. Apart from culture, only a few reports seem to exist that investigated the cells directly from whole blood. Wiegner et al. [182] used standard flow cytometry (FACS Calibur and BD CellQuest software, BD Biosciences, San Diego, CA, USA) sampling 100 uL whole blood. The blood was stained with validated antibody cocktails which was followed by red blood cell lysis and analysis counting in total 3e5 events. Fibroblast-like cells have been isolated from peripheral blood by Epcam positive selection using the CellSearch system giving rise to a CK-/DAPI+/CD45−/vimentin+ phenotype [149] suggesting an intermediate EpCam+/Vimentin+ CD45 negative cell type usually known as the CTC in epithelial mesenchymal transition.

### 5.3. Circulating Fibroblast-Like Cells Clinical Usefulness

Due to inconsistent use of terminology and characterization with respect to hematopoietic and mesenchymal lineage, biomarker-specific conclusions may be difficult to establish. Most potential biomarker applications in cbLB may rely on marker elevation allowing prediction of disease extend and evolution, treatment options or prognostication in various diseases including solid tissue cancer, renal failure, pulmonary hypertension, chronic inflammatory statuses including liver disease [183,184,185] or autoimmunity [139]. Naturally, elevation in fibroblast-like cells would not be very specific to any particular disease.

When it comes to pulmonary fibrosis, Moeller et al. [4] was most cited, having reported elevation of circulating FC in patients with idiopathic pulmonary fibrosis (IPF). The authors suggested a 5% cutoff to predict worse prognosis. The finding of FC elevation in IPF patients was reproduced by Heukel et al. 28 [138]. A positive correlation of circulating FC count has been reported with interstitial lung diseases as a complication of rheumatic arthritis. Higher counts seemed to indicate disease severity and were not associated with rheumatic arthritis alone or C-reactive protein levels [152]. Also, Mehrad et al. [185] showed that circulating FC count and activation state (α-SMA) were associated with abnormal pulmonary function in adults as a complication of sickle cell disease. Moreover, Shipe et al. [186] supported usefulness of circulating FC as to predict asthma severity. As circulating FC elevation is most likely associated with any tissue fibrosis, it is well to expect that cardiac fibrosis may elicit similar findings when compared to pulmonary complications [187,188]. Furthermore, chronic periaortitis patients showed circulating FC elevation when compared to healthy donors, further suggesting predictive biomarker value in fibrotic diseases. FC may well be associated with cancer [189]. However, the field of cancer- associated FC or tissue FC is not well understood [190] and consequently remains to be researched.

Greater biomarker potential may arise from the detection of activated circulating FC so for example FC expressing CD45, Col I and CXCR4 that might indicate abnormality upon detection suggesting active fibrosis, in particular in autoimmune, renal, pulmonary, liver and cardiac diseases [142,143,191,192]. Similarly, the clinical interpretation of in particular circulating myofibroblast (αSMA positive cells) elevation needs more investigation. It can be expected that this marker indicates augmented disease severity. Different functions in repair by fibrocyte-derived myofibroblasts can be assumed, when compared to MSC-derived myofibroblasts [193]. Such investigations would certainly advance cbLB biomarker development. Still unexplored with similarly high potential as (other) activated circulating cells, is the CD45 negative FC. So far, the finding was not related to any human disease [134].

General clinical use of MSC was largely focused on stem cell transplantation for hematopoietic recovery and regenerative treatments. Nevertheless, investigations on biomarker applications have emerged. Wiegner et al. [182] employed MSC quantification in polytrauma patients finding depressed levels in patients when compared to the healthy cohort, which may suggest that elevation is incurred upon chronic or long-lasting disease and deprivation occurs in acute incidences [191]. The oncological biomarker potential of MSC or tumor-associated FB are equally less developed when compared to FC. Of note, fibrotic tumor tissue development may be part of late stage tumor evolution. Therefore, we speculate that tumor-associated FB in the circulation may be associated with later stages of tumor growth and consequently limiting the biomarker to advanced or metastatic disease [194,195,196]. In metastatic prostate cancer, fibroblast-like cells with a phenotype cytokeratin 8/18/19−/DAPI+/CD45−/vimentin+ were detectable specifically in the metastatic setting [149]. The seemingly latest work on tumor-associated circulating FB was presented by Ao et al. [195] that reported clinical relevance with respect to metastasis prediction in breast cancer patients. Given the valuable investigations on fibroblast-like cells in cancer [175,197], we may propose the idea that circulating fibroblast-like cells offer the possibility in cbLB to distinguish tumor states between dormancy and active upon absence or presence of circulating FB and/or MSC, respectively. Also, much needed are markers that discern metastatic/extensive from locally advanced/limited cancers. Perhaps such markers may have been proposed by Jones et al. [148] and Ao et al. [195].

Besides diagnostics, fibroblast-like cells carry potential for use in therapy of various diseases. Therapeutic potentials may be differentiated into pharmacodynamics and regenerative medicine. Kimura et al. [197] investigated the status of fibroblast activation protein (FAP) in FB and suggested that FAP is a target for therapeutic intervention in idiopathic pulmonary fibrosis. Circulating MSC quantification may also be a surrogate biomarker for therapy efficacy monitoring as shown in postmenopausal osteoporotic women treated with intermittent parathyroid Hormone 1–34 [198,199].

MSC based regenerative medicine may profit from several advantages of cbLB that include the retrieval of valuable bio-compatible material from the blood and the potential to replace expensive and invasive procedures such as bone marrow aspiration, or liposuction [200]. Apart from multipotent stem cell properties, MSC were targeted in regenerative medicine due to their homing ability towards sites of inflammation as well as their potency in immunomodulatory and anti-inflammatory effects [154]. Therefore, MSC hold potential as novel approach for curing otherwise difficult diseases that include neural diseases, wounds, MI or various musculoskeletal diseases [142] or several immune mediated diseases, such as Graft versus Host disease, aplastic anemia, Crohn’s disease, rheumatoid arthritis, and multiple sclerosis [154]. In general, all fibroblast-like cells may inherit wound healing properties. FC may stimulate dermal cell proliferation, keratinocyte proliferation with re-epithelialization, and angiogenesis [140,143,193]. Administration of MSC may support the efficiency of bone regeneration, joint repair with respect to tendon injury and skeletal muscle alterations [158] However, the general clinical usefulness of fibroblast-like cells remains to be proven in human disease.

Moreover, fibroblast-like cells are of interest in the science of aging [143,191]. Al Saedi et al. [201] investigated the concentration of circulating osteoprogenitor cells with phenotype CD45+/osteocalcin+ in healthy donors. The authors reported cell deprivation in aged individuals and in particular low lamin A circulating osteo-progenitor cells to be useful as reliable biomarker for frailty at high age [202].

## 6. Very Small Embryonic Stem Cells

### 6.1. Very Small Embryonic Stem Cell General Background

Initially described as tissue committed stem cells, the name has been changed to very small embryonic stem cell (VSELS) due to their small size appearing in diameters of a few micrometers and the identification as an adult tissue-residing in fact pluripotent stem cell with similarities to primordial gonad stem cells [203]. Although having been questioned [204], ongoing research does not seem to shed any doubt about the existence of VSELS in various tissues) [205]. The cells can be found seemingly throughout the body and highest concentrations may be found in the brain, kidneys, muscles, pancreas, and bone marrow. This cell type is equaled to the most primitive stem cell population with capacity to differentiate into cell lines from all three germ layers so for example into mesodermal cardiomyocytes, ectodermal neural cells, or endodermal pancreatic cells. The exact function of VSELS remains to be investigated. VSELS were shown to differentiate into the hematopoietic stem cells and consequently, were suggested to be precursors of long-term repopulating HSC) [205]. More general, VSELS were suggested to be a reserve population of stem cells and tissue-committed progenitor cells, which are mobilized after tissue injury [206].

Similar to other stem or progenitor cells there is no single marker specific for VSELS and was characterized by phenotypes including CD34+, CD133+ Lin− and CD45−. Other markers included CD184+, Oct3/4+, Nanong+, SSEA-1+, Sca-1+. The expression of markers typical for primordial germ cells can be expected and include Stella, Fragilis, Nobox, Rex-1, Hdac6, and CXCR4. The nucleus was described as high fluorescent upon Dapi or Hoechst staining. Human peripheral blood and bone marrow-derived VSELS are significantly smaller in diameter than monocytes and granulocytes, and are larger than platelets. First visualizations have been conducted by Image stream technology [207].

It has been postulated that VSELS are mobilized into the peripheral blood from bone marrow or possibly other organs in response to damage and hypoxia consequently, would home damaged tissue as to repair or regenerate similar to the MSC. Hereby, stromal-derived factor-1 (SDF-1) and other chemotactants were suggested [208]. However, given the relative high concentration (see Section 9), VSELS egress into the circulation may follow a similar steady-state physiological process as described for the HSC. Few reports exist about investigations on VSELS in the circulation of adults and deserves to be rated as neglected rare cell type when compared to the attention given to other circulating stem and progenitor cells) [209].

Upon its first reporting in murine bone marrow, VSELS were identified by Facs sorting selecting phenotypes Lin-/Sca-1+/CD45− and Lin-/Sca-1+/CD45+ cells from nucleated cell suspension after RBC lysis [203]. Ever since, mostly flow cytometry strategies have been employed to identify this cell population in peripheral blood. Moreover, the flow cytometry gating strategy has been adapted taking into account the size of the cells [206]. Peripheral blood pre-enrichment was divided into red blood cell lysis and a specific sequential ficoll centrifugation strategy that resulted in a fraction potentially containing very small nucleated cells referred to as the 5th layer [210]. In this procedure, a histopaque purification system was used extracting the erythrocyte-granulocyte cell layer beneath the Ficoll surface. During further pre-enrichment, RBC were separated from the small cells by stronger centrifugation. The resulting pellet (5th layer) was resuspended in RPMI and the supernatant was discarded. In general, VSELS recovery was reported more efficient using erythrocyte lysis when compared to the “5th layer method” [211]. More recently, clinical investigations on peripheral blood VSELS content followed no other method as in bone marrow or umbicilical cord blood however, having chosen a stain-then-lyse then-wash protocol and using 600 uL of sample blood [212] as opposed to the first lyse-then-stain method. Also, Ratajczak et al. [209] and his group proposed a three step method briefly consisting of red blood cell lysis, followed by positive selection of CD133+cells using immuno -magnetic cell separation technology, and thirdly, FACS-based isolation of small CD133+Lin−CD45− cells. VSELS have been cultivated following the procedure of 206. Gounari et al. [205] isolating VSELS from the fifth layer then further enriching the cells by cultivation using SDF-1a in an alpha-MEM culture medium, collecting CXCR4 + VSELS that migrated toward the SDF-1a factor as the main chemotactic factor. Also, for being CD45 negative, the common CD45 negative depletion assay is well suited for the isolation of VSELS. Figure 5 shows typical very small CD45 negative cells that correspond to descriptions of published imagery. In order to advance knowledge about circulating VSELS, sufficient isolation of such cells is required. Herein, the small size of VSELS is a problem to many cbLB platforms relying on selection by size commonly focusing on larger cells and therefore introducing an unacceptable bias towards this cell type.

### 6.2. VSEL Stem Cell Clinical Usefulness

VSELS have been under investigation as biomarker in various diseases as well as source in stem cell-based therapy. In cancer cbLB, VSELS remain largely unexplored. Seemingly, VSELS may be valuable in contributing to the wealth of diagnostic information yet may seem lower specific towards certain pathologies and elicit a lower signal to noise ratio given the relative high physiological concentration (see Section 9). In contrast, VSELS could turn out to be the most attractive cell target in stem cell therapy. VSELS quantification as such VSELS elevation was shown in various pathologies, including nephropathies, vascular pathologies, MI, pulmonary hypertension, chronic obstructive pulmonary disease, stroke, active inflammatory bowel disease or solid tissue cancer and leukemias. Glomerulonephritis, in particular IgA nephritis has been investigated by Eljaszewicz et al. [212] showing elevation specifically in VSELS when compared to other stem cells when tested in parallel, that included HSC, EPC as well as different monocyte subsets with varying maturation and angiopoietic potential. Also, MI may be associated with VSELS elevation [213,214,215]. Still, the potential use as predictive biomarker for MI remains to be assessed. Wojakowski et al. [215] investigated acute MI in association with VSELS concentration and pointed out that the biomarker may be treated with care for the parallel dependency of concentrations on age obtaining lower counts at higher age and in consequence falsely interpreting good outcome. The age interference with underlying clinical conditions was also reported by Sovalat et al. [216]. Furthermore, VSELS have been associated with stroke [217]. It has been concluded that strokes cause elevation of CXCR4+ VSELS and numbers positively correlated with stroke severity. The actual interesting biomarker application in this case is prognosis of stroke recovery [217]. Herein, VSELS phenotypes that involve the markers CD34, CXCR4, and CD133 were shown to be indicative for best recovery upon detection in posterior circulation infarcts and for early recurrence in partial anterior circulation infarcts. In cancer, in particular leukemias were reported to be associated with VSELS elevation [216]. Diabetes seems to elicit reduced numbers [215]. In consequence of age-dependent concentrations, VSELS may also find application in investigations on aging and lifestyle. Similar to MSC, VSELS seem to gradually decline in quantity and functionality with age in the bone marrow and so mostly likely in the peripheral blood [218]. Interestingly, daily exercise was shown to increase VSELS levels suggesting positive effects in particular with respect to coronary artery diseases [206].

VSELS have undoubted potential for use in regenerative medicine. In mouse studies, application of VSELS was shown to prevent left ventricular dysfunction after MI [219,220]. Therefore, VSELS are investigated in human cardiovascular diseases to promote myocardial recovery or improve cardiac function in MI patients [206,216,221]. Furthermore, VSELS may be useful in stem cell therapies for leukemia, lymphoma, hereditary blood diseases and bone marrow failure) [211]. So far, peripheral blood as a source of VSELS in stem cell therapy may need elaborate investigations as the number of cells may be still too low.

## 7. Epithelial Cells

### 7.1. Epithelial Cell General Background

Epithelial cells (EC) line the outer surface of all organs and the inner surface of cavities in organs as well as being major cell type in glands. EC have gained overwhelming importance in solid tissue cancer cbLB after having found epithelial-like cells in the circulation of carcinoma patients then being referred to as circulating tumor cells [1,222]. However, the cell character as being truly epithelial may stand in question without prior knowledge of origin and without having profoundly investigated the cell character [223]. Furthermore, epithelial CTC are often described as round cells with relative low N/C ratio and consequently, may rather resemble immature blast-like cells thus, are substantially different when compared with tissue-derived mature EC that come with a distinct morphology. Mature EC can be classified according to their morphology into squamous, columnar or cuboidal shapes. Very little is known about circulating epithelial progenitor cells that are most likely common in the CRCP and may be cytokeratin positive as well as expressing the chemokine receptor CXCR4. Similar to other circulating progenitors, a steady state situation might be assumed and their role may involve repair of epithelial tissue upon chemotactic engraftment [224]. Gomperts et al. [224] reported the presence of CK5+CD45+CXCR4+ phenotype in the circulating of mouse. However, the positive CD45 remains in question to correctly define epithelial progenitors as to what stage the cell would lose CD45 expression.

EC and likewise epithelial CTC, stain positive for cell surface markers EpCam (CD326, Figure 6) and/or intra-cellular cytokeratin and negative for CD45 in the presence of a nucleus. Depending on the tissue origin and type of epithelial tissue, different cytokeratins have been identified, but also tissue specific markers, such as Pdx-1 for the pancreas cells [225].

As a gold standard definition of the epithelial CTC given by CellSearch^TM^, CTC are defined as nucleated cells of round or oval morphology, a cell size >4 µm in diameter with a phenotype of EpCAM±/CK+/DAPI+/CD45−. This cell type has been exhaustively described for cancer cbLB and reviewed with respect to phenotyping, isolation, and clinical use [226]. Of note, CD45 counterstaining is mandatory for EC identification as also hematopoietic cells can express epithelial antigens or show rare benign epithelial inclusions ascribed to phagocytation of keratin debris [227,228,229]. In view of the said differences between mature EC, CTC and epithelial progenitors, it is sensible to further divide circulating EC into mature cells of clear epithelial morphology (type 1), the relative small and round “CellSearch” EC (type 2) and circulating epithelial progenitor cells (type 3). Type 1 EC may be often overlooked in cbLB for being rated as contamination as inflicted by phlebotomy.

### 7.2. Circulating Epithelial Cells

There is little evidence to support the notion that type 1 and 2 EC circulate steady-state given the low numbers detectable by current cbLB platforms under physiological conditions. However, reports and our own data may allude to occasional cell findings in healthy donors supporting the idea of “destructive” blood non-native cell types. Herein, a low percentage of cases was reported to be positive for type 2 EC across investigators [222,230]. Non-specificity to malignancy and occurrence in healthy donors has also been investigated and discussed by Rosenbaum et al. [231]. Healthiness is subjective and an expandable terminology. Circulating EC positive individuals may have underlying undetected benign conditions such as low chronic inflammation, hyperplastic polyps, asymptomatic benign cysts or infections and may be subjectively perceived as healthy. In particular, a strong association between benign inflammatory disease and circulating EC has been reported [232,233,234]. Consequently, underlying micro-lesions may contribute to the circulation of epithelial cells in alleged healthy individuals. Circulating epithelial progenitor cells are most likely bone-marrow derived and recruited to contribute to repair for example in lung injury [224]. Even though unreported, we argue that the presence of type 3 ECs is within likelihood in particular in non-physiological conditions and warrants more investigation.

### 7.3. Circulating EC Usefulness

In awareness of at least three types of circulating epithelial cells; the mature tissue-derived epithelial cell, the round epithelial CTC and the circulating epithelial progenitor cell, investigations on usefulness of circulating EC was largely limited to type 2 EC perhaps owned to the importance of the cancer topic [1,222] However, since the introduction of the CellSearch CTC (type 2) as prognostic biomarker, little efforts seemed to have been made in the translation of this marker into pathologies other than cancer. Moreover, advancement with respect to usefulness of type 2 EC in early stage cancer or even pre-malignancy may be qualified as pending. Type 1 and 3 EC can be considered neglected. For cbLB in particular cancer cbLB, the epithelial progenitor cells remain largely uninvestigated. Apart from cancer, it was suggested that autologous circulating epithelial progenitor cells constitute a therapy entity by themselves for pulmonary complications [224].

## 8. Miscellaneous

A few more rare cell types may also be part of the CRCP that include the hemapoietic stem cell, the hemangioblast, cells denoted as CH-cells and the trophoblasts. In contrast to before mentioned cell types, the herein discussed CRC may only share the aspect of rarity rather than and the joint association with pathologies.

The uncommitted quiescent HSCis the archetype bone marrow-derived stem cell and part of the spectrum of peripheral blood CRC. The multipotent HSC correctly denoted as long-term multilineage repopulating stem cell may be sufficiently distinguished from other herein discussed rare cells, yet may be closest to VSELS with respect to size and progenity [214]. Uncommitted HSC were often characterized in bicolor flow cytometry using the CD34 positive and CD38 negative phenotype. Moreover, HSC express CD45 and p glycoprotein and express no or low levels of lineage specific antigens including CD33, CD7, and CD10, Thy1 antigen, CD71, HLA-DR and show a low retention of rhodamine 123. Nevertheless, uncommitted HSC may differ in behavior depending on tissue origin [235].

Undoubtedly, HSCs are in steady state in the circulation. Their egress as such migration into the blood is expected to be regulated by cytokine granulocyte-colony stimulating factor (G-CSF) and chemokines. More recently, Katayama et al. [70] introduced sympathetic nervous system signaling as a new player in the control of HSC migration.

The predominant clinical application of circulating HSC is stem cell transplantation prior to bone marrow mobilization into peripheral blood, for example used in radiotherapy patients for hematopoietic re-constitution [236]. HSC detection and characterization may hold usefulness for example in cancer and non-oncological diseases. However, HSC biomarker potential has been rarely investigated and similar to other rare cells may open new opportunities in biomarker discovery. Zahran et al. [93] investigated HSC behavior in hepatic carcinoma patients for the prediction of treatment outcome. The authors investigated the association of HSC and EPC (CD144+ CD34+ CD133+ CD45−) concluding that abnormality was indicated by a EPC shift from low to high and arguing that HSCs alone are not associated with HCC. However, the study did not comply with the definition of HSC uncommittedness having defined the phenotype CD34+/CD133+/CD45− as HSC that may well overlap with endothelial progenitors or VSELS. Eljaszewicz et al. [212] investigated the regenerative potential of IgA nephritis patients by quantifying the frequencies of several stem cell types in the peripheral blood, amongst them VSELS and HSC. The authors did not find a correlation between disease and HSC count having used the phenotype CD235a-/CD45+/CD133+ cells to describe HSC. In both cases, a proper characterization of uncommitted HSCs was not done, so that the role of the uncomitted HSC in many diseases remains to be investigated.

We adopted thehemangioblast to the spectrum of CRC for being reported to circulate in healthy donor blood [51,237]. Multipotent hemangioblasts are endowed with the ability to differentiate in both hematopoietic and endothelial lineages. Therefore, hemangioblasts are progenitors of different herein discussed rare cells, such as the EPC [238], but also MKC progenitors [15] and erythroid and definitive hematopoietic precursors [239]. The cell was characterized phenotypically as negative for CD45, CD133, and CD34, yet expressing a stem cell profile comprising c-Kit and CXCR4 as well as EphB4, EphB2 [237]. The cells may show morphological similarity to VSELS (Ciraci et al. [237] in Figure 3c). The cells have been identified using a two-step isolation procedure, firstly enriching Lin negative MNC by negative selection and secondly further sorting the cells by FACS. The frequency was reported to measure on average 0.6%. A statement of concentration in whole blood was not given.

To the best of our knowledge only the group around Sicco et al. [240,241] dedicates research to the investigation of the then named circulating healing cells (CH cell) that may represent a separate group of multipotent adult progenitor cells that circulate in steady-state the peripheral blood). The frequency in whole blood was not assessed. So far, little is known about the exact phenotype, yet expression profiles may be closest to MSC. The cells are negative for lin marker and CD45 and may resemble in morphology those of VSELS and are reported to express bone marrow stromal antigen 2 [241].

Trophoblasts are common in the circulation of gravidas as being fetal-derived and as such viewed as an attractive cell type for NIPT. The cell type characterization is problematic due to the lack of specific antibodies and the overlap with MSC resulting in commonly experienced challenges of isolation and in-depth investigations of subtypes. Also, the cells show about 1% incidence of chromosomal mosaicism and may occur in multi-nucleated state which makes FISH analysis of aneuploidies difficult. This cell type can be subdivided into cytotrophoblast, syncytiotrophoblast and extravillous trophoblasts. To the best of our knowledge, reports about trophoblast identification was mostly carried out in maternal blood and only concerned extravillous trophoblasts (EVT). EVTs were isolated by magnetic separation technology targeting the endothelial marker CD105 [242]. Cell frequency in gravidas was reported to measure 5 cells per 20 mL whole blood.

## 9. Circulating Rare Cell Population Concentrations

Knowing the “ground state” concentration of the CRCP is essential to cbLB biomarker development and gives rise to standardized cutoff values that allow objective discrimination between normal and abnormal blood cell profiles. In consequence of standardization and objective determination of abnormality, cbLB may advance towards early stage disease detection. However, it may seem ambitious, if not impossible that one rare cell platform alone enables detection of all possible CRC types given the heterogeneity in physical and immunological properties and varying concentration ranges. Few investigations exist that were purposed to measure the physiological concentration range of certain rare cell types [10,26]. Nevertheless, respective knowledge can be derived from clinical investigations on rare cells that included healthy donor control cohorts. However, the reliability of such data is low due to lack in standardization with respect to technology and cell characterization.

Bone marrow-derived MKC commonly circulate, which has been known already for over five decades [25]. Most useful information about frequencies was provided by Melamed et al. [26] and Hansen et al. [30] that reported an average concentration of naked MKC measuring 5 cells in a range of 0 to 25 cells per mL (excluding rare outliers). So far, no evidence has been brought forth of cytoplasmic MKC in healthy donor peripheral blood. Ascribed to the alleged intra-pulmonary filtration, spine and pelvic girdle draining vena cava blood samples may contain 10× more cytoplasmic MKC than peripheral blood [21,26]. If present, cytoplasmic MKC concentrations may then range far below 0.5 cells/mL in peripheral blood [21]. Recent clinical investigations on rare cells that included the detection of circulating MKC did not seem to reproduce past findings and often re-discovered MKC by accident [11]. Moreover, healthy controls were reported to contain no or very low average concentrations (1 cell per 7.5 mL) [11,29,34]. We speculate that the high fragility of this cell type and inherent technological flaws of employed rare cell platforms are cause of the experienced paucity. Of note is the finding of higher concentrations of CD41 positive/CD45 negative cells by Fachin et al. [2], yet presenting imagery of smaller cells not exceeding 15 µm in diameter with normal round nuclei (shown in supplemented material), leading one to speculate the find of megakaryoblasts instead of MKC. It shall be added that a common problem in CD41 characterization is platelet adhesion that may lead to misinterpretation [243].

Circulating EnC may represent events of frequency between 0.01% and 0.0001% of mono-nuclear cells, which equals to an approximated cell concentration range of 1.5 to 150 cells per mL in healthy donor peripheral blood (assuming a yield of 1.5 × 10^6^ MNC per mL whole blood) Dignat-George et al.) [77,95]. Numbers about circulating EnC and EPC in healthy donors seem to greatly vary between investigators which has been ascribed to marker choices, and more profoundly to quality issues in enrichment and/or analysis technology [94]. Awareness about enumeration inconsistencies and its remedies exist [244]. Torres et al. [75] reported a median number of 504 circulating EnC. CEC were identified as CD45 negative, CD146 positive and CD133 negative cells. Much to the difference, Alessio et al. [76] measured only 100 EnC per mL on average in healthy donors. Circulating EnC were defined as positive events to CD31, CD144, CD146 and negative to CD45 and CD133. Both investigators relied on flow cytometry. As can be expected, analysis by microscopy seems to yield significant lower concentrations [77]. The CellSearch platform was modified to detect circulating EnC, reporting considerable lower concentrations measuring 3 circulating EnC in median number with a max. of 13 cells per mL [78]. Lin et al. [87] employed a rare cell isolation protocol developed by Cytelligen Inc. (San Diego, CA, USA), reporting aneuploidic circulating EnC with phenotype CD31+/CD45−/Vim-/Dapi+ in healthy donors measuring on average 0.5 cells per mL. Bethel et al. [68] relied on a low cell loss RBC lysis-then-staining method applied to peripheral blood samples from MI patients and healthy donors. Therein, healthy donor concentrations of circulating EnC would not exceed 0.5 cells per mL. Also, Bhakdi et al. [34] investigated the EnC content in prostate cancer patients as well as healthy and non-malignant individuals reporting mostly none to a few cells per mL in non-malignancy patients becomes evident that microscopy analysis persistently measures lower concentrations when compared to flow cytometry. Kraan et al. [61] pointed out potential pitfalls in flow cytometry and used a more rigorous analysis strategy, then in fact measuring similar counts as obtained by the CellSearch method when tested in parallel. Hereby a concentration of 1 to 20 circulating EnC per mL were reported. Reports published by Bethel et al. [68] and Lin et al. [87] may come closest as we believe to the truthfulness of a lower physiological concentration range of a few cells per mL.

The majority of reports about circulating EPC concentrations are dominated by flow cytometry analysis and may support the finding of higher concentrations when compared to circulating EnC, suggesting steady-state circulation of bone marrow-derived EPC. However, numbers are not really comparable given the differences in used phenotypes. Hansmann reported on average 140 CD34+/KDR+ EPC per mL peripheral blood [74]. This concentration range may have been reproduced by Schmidt-Lucke et al. [101]. However, the phenotype can hardly stand for the definition of EPC. Rhone et al. [92] also measured higher concentrations in healthy individuals with a median number of 360 cells per mL having used the phenotype CD45−, CD34+, CD133+, CD31+. Torres et al. [75] defined EPC as CD45 low or positive, CD146 positive and CD133 positive cells having measured 294 circulating EPC per mL in healthy controls. A stricter definition as positive events to CD 133, CD 144, CD 146, VEGFR-2, and negative to CD 45 and CD 31 resulted in no findings [76]. Also, Lin et al. [67] applied a strict phenotyping strategy to exclude early EPC and to include endothelial colony-forming cells. This approach yielded in fact very low concentrations when cultivated measuring 0.58 colonies on average per mL. As it may seem, definitions of circulating EPC include several different subtypes as well as overlaps with hematopoietic cells, so for example the CD34+/KDR+ phenotype. Therefore, most EPC definitions are unfit for comparison of concentration levels, thus requiring deeper and more stringent phenotyping. We may conclude that early EPC concentration levels may range between 140 to 360 cells per mL and late EPC concentrations are very low concentrated similar to mature circulating EnC.

According to routine hematology analysis, NRBC ought not to be found in healthy adult individuals [122]. However, provided that efficiency enrichment is efficient, EB can be detected commonly in healthy donor peripheral blood. Our work supports a median concentration of matured and less matured EB in healthy donors of 1.5 cells per mL using the CD71+/GPA+/CD45− phenotype [10]. Larger immature EB were also identified in 47% of donors in range of 0.2 to 1.1 cells per mL. Investigations on CRC types based on the CTC-iChip enrichment platform revealed a NRBC content of 16% of enrichment “left over” cells [2]. A number of at least 500 left over cells per mL whole blood was given and translates into NRBC concentration in whole blood of greater 50 cells/mL. The platform may be more sensitive towards EB when compared to Schreier et al. [10], yet the phenotypical expression excluded the detection of CD71 expression relying on the expression of GPA only. It is suggestive that less specific staining yields more positive events. Of note is that the group presented rather small cells, suggesting that the findings were limited to matured EB and raises the question of the whereabouts of immature EB exceeding 12 µm in diameter. More information about EB concentration ranges may be obtained from investigations on NIPT thereby having quantified fetal as well as maternal EB. However, pregnancy may not correspond to the truly physiological situation. In attempts to cultivate fetal EB, circulating EB colony forming units measured 13.8 cells per 1 × 10^5^ MNC suggesting an amount of in fact 207 vital EB per mL (calculating 1.5 × 10^6^ MNCs per mL whole blood) [115]. This number may be the highest reported. Troeger et al. [118] reported the detection of on average 92 cells/mL in pregnant women with roughly 50% of those being of fetal origin. The percentage of fetal EB may even be slightly higher [119]. Recent contributions may support earlier reported concentration levels measuring on average 49 EB of fetal and maternal origin per mL in pregnant women. A notably lower concentration was reported by Kwon et al. [119] measuring 2.9 circulating EB per mL. Such variations in number may be related to methods in enrichment, yet also staining. Highest counts seem to be produced by the analysis of the GPA+/CD45−/Dapi+ phenotype or Giemsa Gruenwald staining. Given the few data, we suggest a concentration range below 50 cells per mL to be within possibility under physiological conditions.

CD45 negative or truly non-hematopoietic (mesenchymal) circulating fibroblast-like cells represent the lowest and CD45 positive circulating FC represent the highest concentration range of CRC. Circulating FC may not exceed 5000 cells per mL with a median number of 1400 cells per mL in healthy donors (CD45+/CD34+/CD11b+) [138]. Apparently, MSC can be found in the circulation, yet mostly upon disease or mobilization technology. Cultures from mobilized peripheral blood progenitor cells of cancer patients or healthy donors yielded only a few adherent cells which showed fibroblast-like morphology, yet did not form bone in vivo [170]. The commonly reported frequency of MSC when mobilized was reported to be in range of 0.5% to 0.6% relative to all nucleated cells [162,165,166] Past reports concluded that isolation of MSC from non-mobilized peripheral blood was not successful [156,162]. However, this might not be correct. According to Zvaifler et al. [180] circulating MSC shall occur in frequencies of 2 to 10 cells per mL having used healthy donor buffy coat cultures. Most recently, Lin W. [156] reported successful isolation of MSC in 60% of cases from healthy donor peripheral blood relying on red blood cell lysis and subsequent plating at low concentrations. Unfortunately, the study was more qualitative and the authors did not report counts of adherent cells at day 0. Nevertheless, the provided study imagery suggests that MSC are present at relative high numbers (over 1 cell per mL). Similar to observations in other CRC, flow cytometry analysis yielded overall higher concentrations levels of MSC. Wiegner et al. [182] quantified MSC in a healthy donor cohort as baseline for polytrauma patients and measured a mean frequency of 21 MSC per 1e6 cells having used the phenotype CD45−/CD73+/CD90+/Stro-1+ and of 50 cells per 1e6 having used the phenotype CD45−CD105+CD166+STRO1+. Sielatycka et al. [245] reported even 100 to 1000 cells in pregnant women using the CD45−/CD105+/CD90+/CD29+ phenotype. A concentration that exceeds 50 cells per mL seems unrealistic given the paucity and rarity of adherent cells suggesting that either most mesenchymal lineage fibroblast-like cells are non-adherent or including the possibility of analytical flaw or overlapping phenotypes [94,162]. We would also like to point out the issue of excluding the analysis of the cell nucleus in most flow cytometry methods. Given the few reports and the high concentration range differences by flow cytometry, the cultivation data as reported by Zvaifler et al. [180] seem more reliable for our estimation then suggesting a physiological concentration range of 2–10 cells per mL.

VSELS may come second to circulating FC in abundance, reporting frequencies in adult peripheral blood in range of 0 to 1500 cells/mL relying on flow cytometry [206,215,217]. According to Eljaszewicz et al. [212], maximum VSELS concentration levels in human adult peripheral blood are much lower, measuring roughly 0 to 350 cell per mL with in fact no positive findings in some of the donors. Different concentration levels may be ascribed to differences in gating strategies. Eljaszewicz et al. [212] established a VSELS specific gating strategy that would exclude large cells using the lin-/CD235a-/CD45−/CD133+. In murine peripheral blood, Kucia et al. [208] measured roughly 100 to 200 VSELS per mL which supports the finding of Eljaszwicz [212]. In mobilized blood (granulocyte colony-stimulating factor), VSELS quantity measured 800 cells/mL, confirming the assumption of bone marrow origin [216]. In conclusion, the physiological concentration VSELS may be in range of 0 to 350 cells per mL when taking into account sensitive flow cytometry gating strategies.

The cbLB breakthrough publication as authored by Allard et al. in [1] claimed that circulating EC defined as CK+/CD45−/Dapi+ round larger blastoid cells are specific to solid tissue cancers, suggesting that circulating EC are uncommon in healthy donors. Nevertheless, this notion has been refuted [230,246]. Lustberg et al. [230] reported circulating EC quantities up to 6.7 cells per mL in healthy controls using the putative definition of CK+, EpCam- CD45 negative cells. The cell definition with respect to CK or Epcam seems to affect cell numbers as a significant drop in counts can be expected upon addition of EpCam surface marker expression to the phenotype, then measuring 0.4 EC per mL. Ozkumur et al. [247] reported a similar range with a maximum number of 0.7 per mL and a median number of 0.19 cells per mL using a similar type of analysis, when compared to Lustberg et al. [230]. In congruence, Tsai et al. [248] investigated the prognostic quality of circulating epithelial cells for metastatic colorectal cancer thereby investing colon polyps of various stages and healthy individuals. The reported mean counts for healthy individuals without polyps measured 0.25 cells per mL and for those with hyperplastic polyps was increased measuring 0.65 cells per mL. Rhim et al. [249] tested healthy controls in comparison to pancreas lesions and reported on average 0.3 cells per mL in roughly 16% of the healthy donors. Though not really healthy, Hardingham et al. [233] reported circulating EC in benign bowel inflammation conditions in 12% of the cases. As a conclusion, epithelial cells may neither be common to the CRCP nor blood native. Yet concentrations can be expected to measure below 1 cell per mL with 0.42 cell per mL on average taking into account the herein cited publications.

In contrast to any other rare cell type, stringent standardization efforts of HSC quantification was pursued and enabled clinical use in stem cell transplantation therapy [250,251,252,253,254]. It is said that 1% of bone marrow cells and 0.1% of peripheral blood MNCs are CD34+. In peripheral blood the frequency might range between 0.01% to 0.05% [251]. As can be expected, the uncommitted HSC fraction may measure a third to a tenth within the CD34+ population [250], then calculating a frequency of 0.003% to 0.017% or a minimum of 180 up to 1024 cells per mL.

## 10. Concluding Remarks

The greater extend of a population of CRC under physiological conditions in each healthy individual may be evident from this review (Table 2). It shall be noted that the herein given list of rare cell types is still not exhaustive. Awareness of the CRCP and its use in cbLB is increasing and may represent the next generation of cancer cbLB that becomes in particular beneficial in combination with whole genome sequencing applications. It is therefore the intention of this review to inform investigators in the field about possible cell findings and biomarker potentials.

Presumably ascribed to research efficiency, technical specialization and/or maintaining research group profiles, a common approach in cbLB biomarker translation has been to investigate a few if not one biomarker for a certain disease at a time. So for example done using endothelial and epithelial cells in cancer prognosis. What remains in obscurity is the association between disease type and alterations of the CRCP. One or two biomarkers may not achieve high diagnostic performance, yet in view of an entire spectrum of CRC acting and reacting as part of a disease, greater diagnostic potential can be expected. It is well to say that CRCP based liquid biopsy supports prediction, diagnosis, prognosis and therapy at least for the big killers in our times. Therefore, a paradigm shift in cbLB biomarker translation could be the simultaneous investigation of an exhausting reservoir of CRC types for the qualification of one particular disease. This approach may be appropriate and perhaps necessary for highly complex pathologies, such as cancer.

It is suggestive that matured cells in the circulation are result of destructive processes and can be explained in healthy individuals by the assumption of destruction and maintenance as part of the normal physiology. The blood circulation is provider of oxygen and nutrients, yet similarly important a drain of organs for removal of old or loose cells or cell material and metabolic waste products. Therefore, cell homeostasis may be one cause of mature non-hematopoietic cells in the circulation. In contrast, the circulation of bone marrow-derived progenitor and stem cells is a physiological process leading to steady-state peripheral blood concentrations suggesting active traffic from the bone marrow into the peripheral blood and perhaps vice versa. It may then sound plausible, that bone marrow-derived progenitor and stem cells are higher concentrated than mature rare cells.

We learned that many if not most rare cell types seem to overlap with one and the same disease (Table 2), so for example, MI involves alterations of mature EnC, EPC, VSELS, MSC, FB, FC and NRBC. The same may account for solid tissue cancers. As an advantage, different aspects of information to the pathology are provided by different CRC arguing in favor of higher diagnostic potential of the CRCP. As a disadvantage, low specificity to a certain disease limits the use of one CRC type as single independent biomarker. Biomarker potential is further determined by the quality of rarity and means that the rarer a cell type is, the lower the threshold cutoff for pathologies can be set and the earlier a diagnosis can be made. In view of previous findings and contrary to occasional perceptions of rare cell paucity, most of the herein discussed rare cell types are found commonly in the blood. Aspects of rarity include concentration range and the median number. In numbers, concentrations from below 1 cell up to 350 cell per mL representing a lower concentration range and from 350 to 5000 cells per mL representing a higher concentration range. VSELS, FC and HSC account for high concentrated populations thus, weakening potential as diagnostic biomarker and EC and cytoplasmic MKC account for lowest concentrated populations with a presumed median number of below 1 cell/mL, thus allowing low cutoff values.

Still, biomarker development of the CRCP is in its infancies requiring characterization, independent verification and standardization. More knowledge may support much needed standardization of cutoff thresholds for the same biomarker (phenotype) and disease. For example, circulating EnC cutoff values for use in cancer prognosis varied from not using any cutoff [253]. It may not be wrong to say that the more insensitive the technology, the lower are the cutoffs. It has become obvious that recent technology may not always be sensitive enough to detect most common rare cells, as for example seen in investigations on circulating MKC. One may argue that in the know of the physiological concentration range of a certain CRC type, individual cutoffs are obsolete yet instead, technologies that tap into CRCP should be benchmarked on the physiological concentration range in the first place giving rise to a standardized cutoff value. A technical advantage is to avoid validation experimentation using highly artificial cell lines known to be outliers in the parameter spectrum of normal and abnormal blood cells. Another advantage is to avoid non-reproducible cell line conditions in the intra and inter-laboratory setting. In this context, insensitive technology would be barred from biomarker development investigating cell deprivation, for example in coronary heart diseases where the risk of death is associated with lower levels of circulating EPC. Also, insensitive technology is precluded from detecting minor variations in the physiological concentration range giving rise to information about lifestyle, age and health in general by almost any rare cell type. Hereby, mostly stem cells, such as VSELS and MSC have been found to be inversely correlated with age, but also correlated with lifestyle such as smoking and exercise.

Information about physiological concentration ranges depended mostly on flow cytometry. One problem is the often encountered incomparability due to different phenotyping and gating strategies. This may concern overlaps for example between early and late EPC, the EPC and MSC or FC and FB. Standardization efforts in the quantification of rare cells does not seem to exist, apart from the EC (type 2) and HSC [249]. Also, a great discrepancy in cell counts of healthy individuals commonly exists between flow cytometry and microscopy measurements. Investigators using flow cytometry were often aware of the pitfalls of this technology in particular for rare cell analysis [59,95,249]. Moreover, in microscopy analysis, the presence of a nucleus as visualized by various staining methods is mandatory as to identify cells as cells. Nevertheless, this criterion does not seem to apply to flow cytometry analysis. In support of the “nucleus problem”, studies specifically included the cell nuclei into their gating strategy then reporting lower cell concentrations similar to microscopy. Herein, the reports from Hansman [74] and Kraan et al. [61] may be compared. Of note is that according to Mund et al. [72], erroneous counting in flow cytometry was attributed to false positive measurements having detected extra cellular vesicles.

Apart from hematopoietic rare cells, such as FC and the HSC, the majority of rare cells does not seem to express CD45. Technology to access the CRCP in its total appearance in each individual is required to be non-biased in cell selection strategies. The preferred cell selection strategy may then include debulking of red blood cells and depletion of leukocytes, respectively. This has been realized in various forms of magnetic cell separation technology and often referred to as CD45negative depletion assay) [2,10,11].

In conclusion, it may be well to say that past and ongoing investigations have only scratched the surface of the potential of CRC in diagnostics and regenerative medicine. Given the greater diversity of possible antigens, standardized and correct phenotyping of rare cells seems currently problematic. We may observe that current technology is unfit for investigations on the CRCP with respect to interpretation power, loss of information, technological simplicity as well as transferability and costs.

## Figures and Tables

**Figure 1 cells-09-00790-f001:**
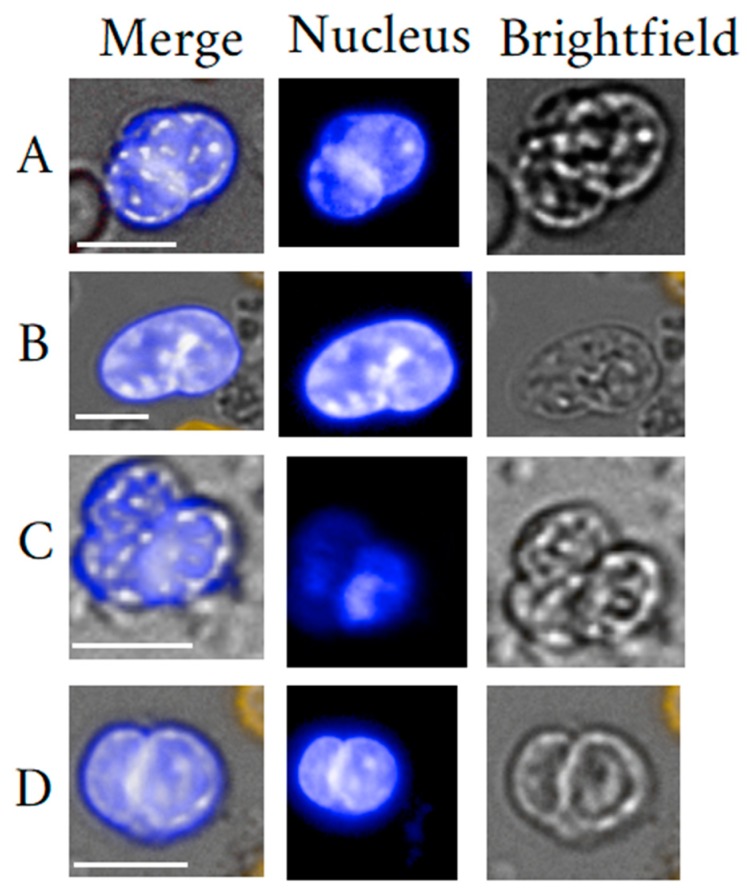
CD45 negative megakaryocyte -like cells detected in healthy individuals by our group from three different donors. Appearance, highest N/C ratio, size as well as shape may allude to the naked MKC. The cells (**A**–**D**) range in size from 14.2 µm to 22.5 µm. White scale bar is 10 µm.

**Figure 2 cells-09-00790-f002:**
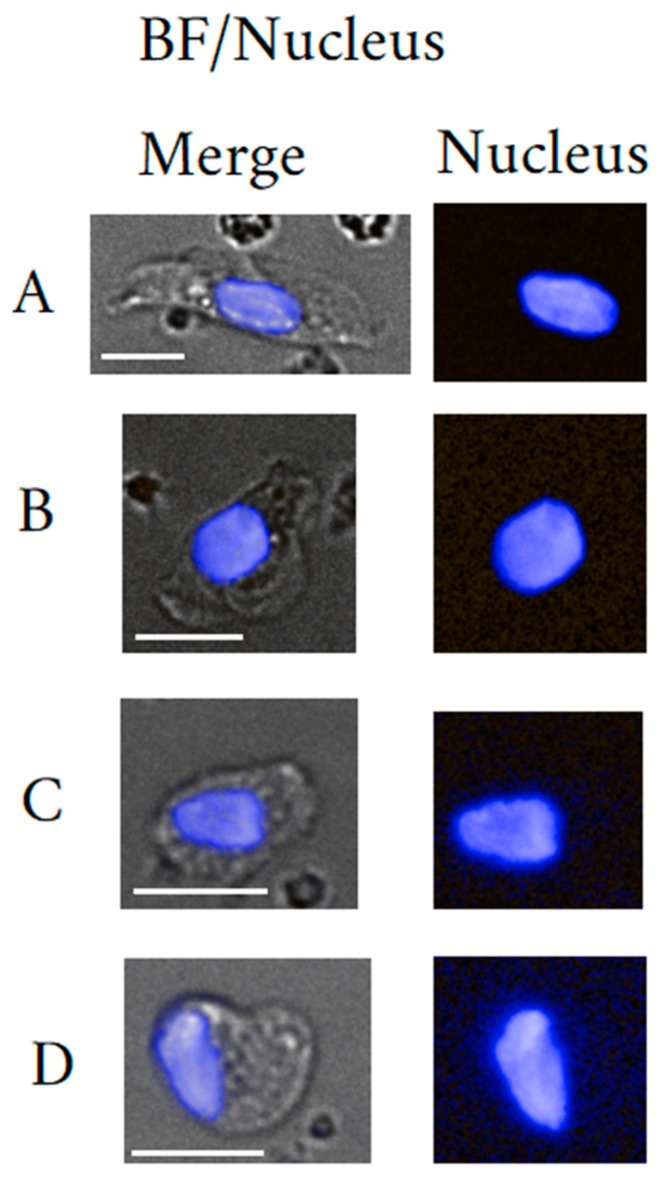
Collection of morphological different mature endothelial-like cells from one healthy donor. Cells (**A**,**B**) represent large cells with low N/C ratio and cytoplasmic pleomorphism. Cells (**A**,**B**) measures 34.5 µm in length 21.4 µm, respectively. This endothelial-like cell type has similarities to activated tip cells. Cells (**C**,**D**) represent smaller cobblestone-like cells with higher N/C-ratio and measure 10.4 µm and 10.7 µm in length (largest diameter), respectively. This endothelial-like cell has similarities to quiescent phalanx cells. The nuclei of all cells show similarities in relative high density, morphology and shape with a size ranging between 7.4 µm to 9.4 µm. The nuclei chromatin structure is amorph and high in fluorescence intensity. White scale bar is 10 µm.

**Figure 3 cells-09-00790-f003:**
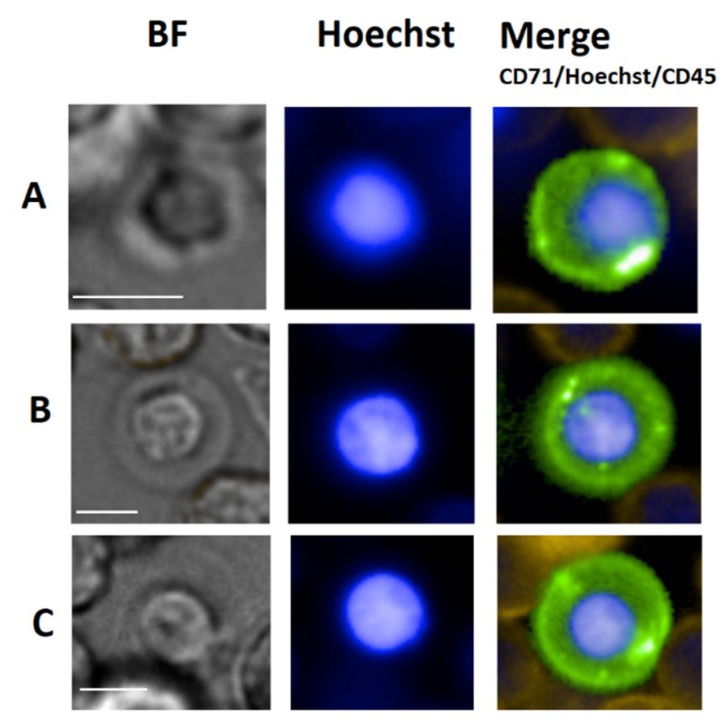
Collection of morphological different CD71+(green)/CD45−(yellow)/Hoechst+(blue) erythroblast-like cells from healthy donors. Morphology and phenotype are characteristic for erythroblasts. Different stages of maturation may be indicated by the nucleus appearance under brightfield as well as high intensity Hoechst signal and the cell size. Cells (**A**) is the smallest and most mature measuring 8.4 µm in diameter, Cell (**B**) is larger and less mature measuring 11.2 µm in diameter, Cell (**C**) is the largest and less mature measuring 13.4 µm in diameter. White scale bar is 5 µm.

**Figure 4 cells-09-00790-f004:**
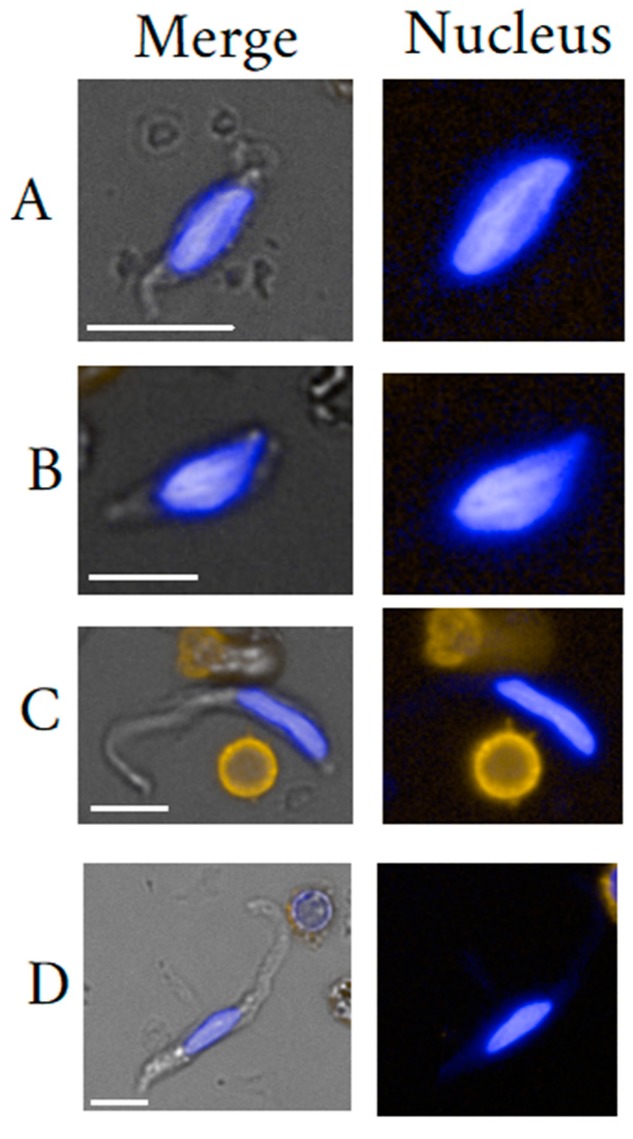
Selection of fibroblast-like cells detected in one healthy donor. Cells (**A**,**B**) are spindle-like and 15 to 18.4 µm in length with thick nucleus of thread-like morphology in contrast to cells, (**C**,**D**) showing very elongated and thin cytoplasm in length of 32 µm to 42 µm and respective thin elongated nuclei, suggesting different cell types or statuses, when compared to cells (**A**,**B**). Cells (**A**,**B**) may resemble descriptions of fibroblasts and cells (**C**,**D**) may resemble descriptions of inactivated fibrocytes. All cells are CD45 negative (otherwise yellow). White scale bar is 10 µm.

**Figure 5 cells-09-00790-f005:**
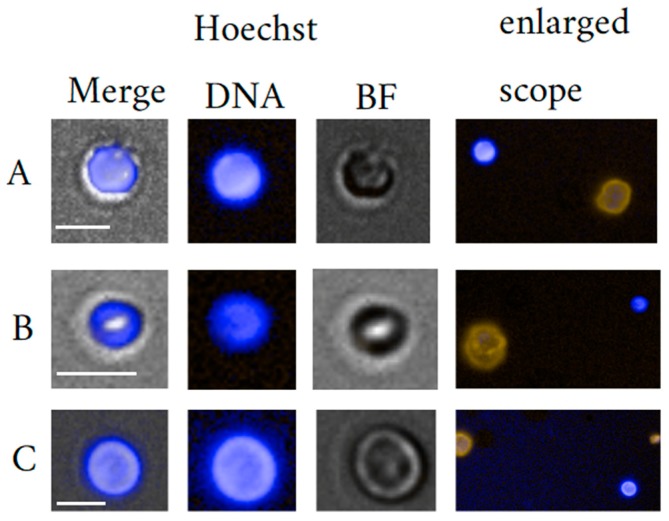
VSELS-like cells with slightly different appearances between each other derived from one healthy donor. Cell (**A**) represents a large VSELS-like cell with 6.7 µm in diameter. Cell (**B**) is smallest with 4.9 µm in diameter. Cell (**C**) represents a medium sized VSELS with 5.7µm. In all cases, the cells exhibit a near 1 N/C ratio and high nucleus staining intensity when compared to leukocytes. White scale bar is 5 µm.

**Figure 6 cells-09-00790-f006:**
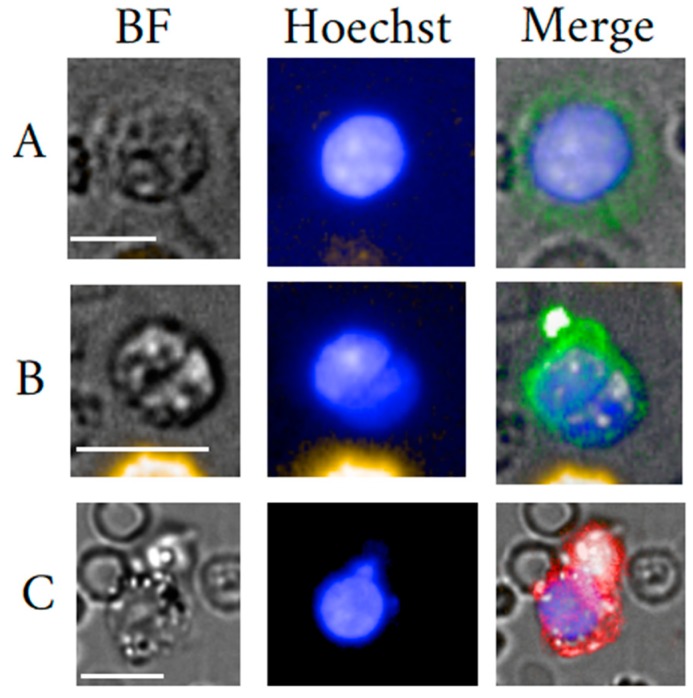
Depicted cells (**A**–**C**) are of phenotype CD326+ (green and red emission)/CD45− (yellow emission)/Dapi+(blue emission). Picture (**A**) depicts a CTC with classic round and low N/C ratio appearance of 12.2 µm in diameter, derived from a metastatic small cell lung carcinoma patient. Picture (**B**) depicts a non-classical CTC from the same patient measuring 9 µm in largest diameter. Picture (**C**) depicts a with a low N/C ratio epithelial cell with an epithelial nose and in size of 18.3 µm derived from a healthy donor. White scale bar is 10 µm.

**Table 1 cells-09-00790-t001:** Suggested MKC biomarkers.

Cellular Marker Characteristic	Quality	Indication/Application
MEP	Present and cultivatable	Platelet therapy
Cytoplasmic MKC (cMKC)	Elevation	Myeloproliferative neoplasms
Proliferative cMKC	Elevation	Essential thrombocythaemia
Pro-apoptotic impaired cMKC	Elevation	Myelofibrosis (MF)
Naked MKC	Elevation	mCRPC good prognosis
Polyploidy	High ploidy >8N	Prediction of metastasis
Thrombospondin-1	Marker expression	Lowered tumor progression
Lysyl oxidase positive cMKC	Elevation	Fibrosis myeloproliferative disorders

**Table 2 cells-09-00790-t002:** List of circulating rare cells.

CRC Type	Description	Cell Concentration per mL	Clinical Usefulness (cbLB)	References
Megakaryocyte-Erythrocyte Progenitors	Bone marrow dwelling megakaryocyte progenitor	N.A.	Therapy: thrombocytopenia	[46,48,49]
Naked Megakaryocyte	Large bare lobulated nuclei with high density DNA	<25	(I) solid tissue cancer prognosis	[11]
Cytoplasmic Megakaryocyte	Largest circulating round cell being original megakaryocyte containing platelet load	<0.5	(I) Therapy: thrombocytopenia (ii) Diagnostic biomarker and therapy intervention target in myeloproliferative disorders (iii) prediction of bone metastasis	[17,20,21,46,48,49]
Mature Endothelial phalanx cell	Cobblestone-like quiescent cells	<100 cells (0.5 to 3 on average) *	(I) Prognosis, Predictive biomarker Solid tissue cancer by marker elevation	[34,68,76,94]
Mature Endothelial tip cell	Larger activated cell status			
Mature Endothelial sprout cell	Larger, activated cell status			
Endothelial progenitor cell	Bone marrow-derived	140–360 (early EPC) <1 (late EPC) **	(I) therapy: coronaryartery disease, neo or re vascularization (ii) prediction biomarker myocardial infarction, pulmonary hypertension and diabetes II, atherosclerotic disease progression etc.	[68,74,94,100]
Erythroblasts -Normoblast	Small late matured erythroblast	<50	(I) Predictive biomarker for leukemia (ii) Prediction of death in critical ill patients	[2,118,122]
Erythroblast - Baso-Eb, Poly-Eb	Larger cells with lower N/C ratio	<0.5		[10]
Fibroblast like cells/Fibrocytes (CD45+)	Rare elongated spindle-shaped leukocyte	<5000	(I) prediction of pulmonary fibrosis	[138,150]
Fibroblast like cells/Fibrocytes (CD45−)	Rare elongated spindle-shaped fibrocyte subpopulation (activated?)	250 ***	unknown	[134]
Fibroblast-like cells/myofibroblast (hematopoietic lineage)	Activated fibrocyte differentiated into tissue resident contractile cell type	unknown	(I) predictive biomarker myocardial infarction	[186]
Fibroblast-like cells/mesenchymal stem cells	Bone marrow derived cell	<10	(I) Prognosis and predictive biomarker in many pathologies e.g., cancer, polytrauma (II) fetal cell marker	[181,183]
Fibroblast-like cells/myofibroblast (mesenchymal lineage)	Activated MSC or fibroblast differentiated into tissue resident contractile cell type	unknown	(I) predictive biomarker for solid tissue cancer (ii) therapeutic target in cancer (iii) stem cell therapy for bone and cartilage repair	[194,198]
Hematopoietic stem cell	Uncommitted quiescent small blastoid cells type	<1000	To be researched	[212,224]
Very small embryonic stem cells	Smallest rare cell subset with high density nucleus	<350	(I) Predictive marker diaease non-specific (ii) age/lifestyle marker (iii) stem cell therapy	[203]
Mature Epithelial cells type 1	Large squamous, columnar, cuboidal shaped cells with very low N/C-ratio	Unknown	Not investigated	
Mature epithelial cell type 2	Round or oval blastoid like cells with low N/C ratio= “CellSearch CTC”	0.42 cells (average of reports)	(i) prognosis, prediction in solid tissue cancer	[1]
Circulating epithelial progenitor cells	Round blastoid cells	Unknown	Pulmonary disorder	[146]
Hemangioblast	Small blastoid round cells (similar to VSELS)	Unknown	Not specified	[237]
CH-Cells	Small blastoid cells	Unknown	Not specified	[241]
extravillous trophoblasts	Large irregular shaped blastoid cells	<0.5 cells	Fetal origin	[242]

* no distinction between activated and quiescent mature endothelial cells, and microscopy range in brackets; ** estimated from cultivating cells reported by Lin et al. [87]; *** cell type unknown in peripheral blood concentration is estimated from report about ratio between CD45+ and CD45− fibrocytes by Suga et al. [134] reporting 5% frequency when referred to CD45+ fibrocytes.

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
