# Peer review of "The Blood Circulating Rare Cell Population. What Is It and What Is It Good for?"

_cells, 2020, doi:10.3390/cells9040790_

Round 1

Reviewer 1 Report

This review is well organized and written in a logical manner. The authors provided detailed background and research information from >200 literatures.

Some minor comments:

1. Figure 1 seems missing;

2. Where is Table 2;

3. FC and MSC could be a subtitle under the title of "5. Fibroblast-like cells";

4. The whole manuscript needs to be reviewed and revised by a native English speaker.

Author Response

Cells-715848 - Response to the reviewer

The Blood Circulating Rare Cell Population. What is it and what is it good for?

Stefan Schreier

Wannapong Triampo

Reviewer 1 – Minor Revision

1. Figure 1 seems missing;

Authors’ response: resolved

2. Where is Table 2;

Authors’ response: resolved

3. FC and MSC could be a subtitle under the title of "5. Fibroblast-like cells";

Authors’ response: complied

4. The whole manuscript needs to be reviewed and revised by a native English speaker.

Authors’ response: improved

Reviewer 2 Report

Schreier and Triampo proposed an exhaustive review article on the existing circulating rare cells in the bloodstream describing in detail all the morpho-physiological characteristics. However, reading the manuscript is very difficult because each chapter is too long and contains information that is of little use in establishing the clinical significance of the circulating rare cells. Furthermore, the manuscript is full of grammar errors and seems to be not fully edited by the authors. Below are suggest some major revisions that the authors have to address before reconsidering the manuscript for publication in Cells:

1) I strongly suggest to the authors to summarize the information contained in the entire manuscript in order to simplify the reading of the text;

2) I do not agree with the choice of the authors. The authors state that CTCs and HSC were not described because are non-physiological condition or extensively described in other reviews. As the title is “The Blood Circulating Rare Cell Population” and the review describe the clinical usefulness of such rare cells, I strongly believe that CTCs should be described in a separate chapter. Same comments for HSC;

3) Figure 1 is not reported in the manuscript;

4) In each Figure, the panels are too small;

5) The review article was not conceived very well. The manuscript presented by the authors seems rather an extensive book chapter describing the various circulating cell populations. This creates confusion in the reader. In addition,  find potentially useful information for the reader along the text is very difficult. The authors describe in detail all aspects of circulating rare cells, however, it is never easy to collect the key information for each cell type described;

6) The majority of References date back to more than 20 years. I think that this review does not add any novelty to the current knowledge on CRCs compared to other similar reviews;

7) What is the usefulness of reporting the circulating rare cells concentration for each cell type? Consider to prepare a single paragraph where the concentrations of all CRCs were briefly described

8) A brief description of the pluripotent common ancestor of all CRCs should be added (with a representative figure too);

9) Chapter 11 is complete and exhaustive. In this chapter no unnecessary information was provided therefore, the readers can easily read it. The authors should summarize all the other chapter as previously stated;

10) In all microscope images provided in Figures, please add the measuring bar;

11) The manuscript is full of English errors. I strongly suggest an extensive language revision performed by an English native speaker. Please correct all the errors (e.g. “in deed” instead of “indeed”; “Vena carva” instead of “Vena cava”; “to there location” instead of “to their location”; “thalassimia” instead of “thalassemia”; “apperatuses” instead of “apparatuses”; “leukoemia” instead of “leukemia”; “indiopathic” instead of “idiopathic”; etc.)

Author Response

Cells-715848 - Response to the reviewer

The Blood Circulating Rare Cell Population. What is it and what is it good for?

Stefan Schreier

Wannapong Triampo

Reviewer 2 - Major Revision

1) I strongly suggest to the authors to summarize the information contained in the entire manuscript in order to simplify the reading of the text;

Authors’ response: A summary has been provided, currently not yet embedded in the text.

2) I do not agree with the choice of the authors. The authors state that CTCs and HSC were not described because are non-physiological condition or extensively described in other reviews. As the title is “The Blood Circulating Rare Cell Population” and the review describe the clinical usefulness of such rare cells, I strongly believe that CTCs should be described in a separate chapter. Same comments for HSC;

Authors’ response: We agree with the reviewers’ opinion. However, it is the intention of this review to elaborate on a cell population that exists under physiological conditions then excluding cells with severe mutational aberrations. The rationale of this restriction is to be able to relate a pathological status of a rare cell to its physiological status with respect to quantity and cell character, consequently requiring knowledge about the situation under physiological conditions in the first place and so intended to provide in this review. In other words, interpretation of abnormality is only possible in the know of the rare cell profile under physiological conditions.

When it comes to CTCs, we have the problem that there is no real physiological status following strictly the definitions of required driver mutations. Furthermore, CTCs are often described as epithelial cells by way of expression of EpCam or CK, which is merely an assumption of its origin. In particular, the positive EpCam status of the EpCam+/CK-/Dapi+ CTC might be acquired along the way thus, leaving the true nature of this CTC type open for speculations. In this view, respective investigations validating biomarker specificity using healthy donor and non-cancer control groups are in danger of comparing apples with pears. What is certainly validated is the specificity of the anti-Epcam antibody in use however, that may not be transcended to a cell type. To cut the long story short, we would to refrain from including CTCs as part of the physiological rare cell population. Nevertheless, the importance of CTCs in rare cell analysis is unquestioned so that we have briefly discussed this biomarker in the chapter 7 (epithelial cells).

3) Figure 1 is not reported in the manuscript;

Authors’ response: complied

4) In each Figure, the panels are too small;

Authors’ response: picture modification will be done accordingly once, the manuscript evaluation is upgraded to minor review.

5) The review article was not conceived very well. The manuscript presented by the authors seems rather an extensive book chapter describing the various circulating cell populations. This creates confusion in the reader. In addition, find potentially useful information for the reader along the text is very difficult. The authors describe in detail all aspects of circulating rare cells, however, it is never easy to collect the key information for each cell type described;

Authors’ response: We are very thankful for this feedback and aware of the crucial issue of the usefulness of this work. We have tried to improve the organization of the text in particular “clinical usefulness”. Major changes in text have been made for megakaryocytes, endothelial cells and fibroblast-like cells.

6) The majority of References date back to more than 20 years. I think that this review does not add any novelty to the current knowledge on CRCs compared to other similar reviews;

Authors’ response: We have remedied this shortcoming. We seemed to have missed a few recent key papers yet, there is no change in the overall comprehension of the rare cell population. Regarding academic novelty, the individual cell types were discussed under different perspectives. Investigations were largely isolated often reviewing one cell type or even subtype in the framework of one specific disease or enrichment/analysis technology. However, the bigger diagnostic picture of the rare cell population is missed. It certainly holds some degree of academic novelty to comprehend this largely separated field in the framework the rare cell population, In fact, awareness of such a population is low amongst investigators which can be observed by reports stating accidental findings of rare cells without relating to alleged existing knowledge.

7) What is the usefulness of reporting the circulating rare cells concentration for each cell type? Consider to prepare a single paragraph where the concentrations of all CRCs were briefly described;

Authors’ response: It is one of our core intentions of this review to comprehend and justify true rare cell concentration ranges under physiological conditions. In fact, we believe that our investigation is still not elaborate enough. This is due to a technical and a diagnostic aspect.

(I) Currently, technology developers in particular enrichment technology have no interlaboratory benchmark apart from head to head performance comparison. This is expensive and often impossible for small research groups or companies. Therefore, spiking experiments are still in use and often tweaked to present good enrichment results. Strangely enough, when putting the proclaimed validated high quality enrichment technologies to the test, old data (e.g. naked MKC) cannot be reproduced and sensitivies remain extremely low for early stage diseases. In the know and wide acceptance of the physiological concentration of individual rare cells, developers are need to reproduce limit of blank values of certain target cells as otherwise deemed insensitive. The establishment of this new benchmark will help to break out of the technology stagnation problem.

(ii) In the know of widely accepted physiological rare cell concentration range, diagnostic assay developers would face a situation with a fixed abnormality cut-off which is opposed to the current situation in cbLB. We have discussed this problem in the concluding remarks that such cut-offs substantially differ between investigating groups, yet using the same biomarker. The reasons is (again) insensitive or incomparable technology and defining lower cut-offs is sign for weak performing diagnostic assays. To us, one big problem of cbLB to inhibit advancement is the ignorance or unawareness of the physiological rare cell concentration.

8) A brief description of the pluripotent common ancestor of all CRCs should be added (with a representative figure too);

Authors’s repsonse: if allowed, we would like to deeper investigate this suggestion and comply within a minor revision. As it currently may seem, VSELs could be potentially hold highest progenity of all CRCs.

9) Chapter 11 is complete and exhaustive. In this chapter no unnecessary information was provided therefore, the readers can easily read it. The authors should summarize all the other chapter as previously stated;

Authors’ response: A summary is provided

10) In all microscope images provided in Figures, please add the measuring bar;

Authors’s response: we would like to comply with the reviewers suggestions within a minor revision.

11) The manuscript is full of English errors. I strongly suggest an extensive language revision performed by an English native speaker. Please correct all the errors (e.g. “in deed” instead of “indeed”; “Vena carva” instead of “Vena cava”; “to there location” instead of “to their location”;

thalassimia” instead of “thalassemia”; “apperatuses” instead of “apparatuses”; “leukoemia” instead of “leukemia”; “indiopathic” instead of “idiopathic”; etc.)

Authors’s repsone: We have sought to improve.

Round 2

Reviewer 2 Report

The authors re-submitted the manuscript almost completely unchanged. The manuscript still needs English editing performed by an English native speaker, this is mandatory. Almost none of my comments have been addressed, but only postponed to the next review process. Therefore my opinion remains unchanged. The manuscript should be reconsidered after major revisions.

I agree with all the authors' replies, except for the comments 1, 4, 7, 8, 9, 10

1) In particular, in my previous comments 1 and 9, I did not ask to provide a summary or an index, but to reduce the length of the text by providing only the key information and not unnecessary and redundant data. So I ask again to summarize and describe the whole manuscript much more concisely and briefly. This would make it easier for readers to understand the manuscript;

2) Please resubmit the manuscript with the new Figures. For a review article also figures should be precisely evaluated (previous comments 4 and 10);

3) Previous Comment 7: I understand the importance of describing the rare cell concentration, however, in my comments I asked to replace the information of all the CRC concentrations in a new separate paragraph;

4) Previous Comment 8: Please add a new brief chapter describing the pluripotent common ancestors of CRCs;

5) The manuscript still must be edited by an English native speaker

Author Response

Cells-715848 - Response to the reviewer

The Blood Circulating Rare Cell Population. What is it and what is it good for?

Stefan Schreier

Wannapong Triampo

Reviewer 2  - Major Revision, 2nd Round

1) In particular, in my previous comments 1 and 9, I did not ask to provide a summary or an index, but to reduce the length of the text by providing only the key information and not unnecessary and redundant data. So I ask again to summarize and describe the whole manuscript much more concisely and briefly. This would make it easier for readers to understand the manuscript;

Response: We agree that the linguistic quality was not satisfactory. We have shortened the text with respect to redundancy. Nevertheless, the qualification of information deemed unnecessary may be debatable. We deleted as much as we thought to be appropriate for the purpose of this review and hope to generated a more concise text.

2) Please resubmit the manuscript with the new Figures. For a review article also figures should be precisely evaluated (previous comments 4 and 10);

Response: The figures have been modified accordingly. Figure 3 (erythroblasts) has been replaced. The reason is that the old figure was copied from our former publication. The journal did not yet responded as to accept publication of this picture. Also, we would like to adhere to the style of the other figures.

3) Previous Comment 7: I understand the importance of describing the rare cell concentration, however, in my comments I asked to replace the information of all the CRC concentrations in a new separate paragraph;

Response: We agree with the reviewer and believe that the separation supports manuscript quality. The information about CRC concentrations as contained in a separate section in each CRC chapter has been extracted and merged in a new separate chapter (chapter 9).

4) Previous Comment 8: Please add a new brief chapter describing the pluripotent common ancestors of CRCs;

Response: Unfortunately, we are still a bit confused about this comment. Also, our knowledge about stem cell lineages is limited. As we understand, pluripotentiality is used to describe stem cell differentiation progeny into all three germ layers and relates basically to embryonic stem cells. Therefore, basically all rare cell type trace back to the pluripotent embyonic stem cell.

Perhaps, we may comply with the following brief elaborations on rare cell ancestry in assumption of multipotentiality;

The hemangioblast seems to be the common ancestor of the endothelial cells, megakaryocyte, erythrblasts fibrocytes, and hematopoietic stem cells. The mesenchymoangioblast was mentioned as earliest precursor of mesenchymal stem cells and so including the fibroblasts. VSELS are claimed to be pluripotent as such are progeny of the embryonic stem cell. Epithelial cells at least those with clear epithelial background can be traced back to the endodermal progenitor cell that does not seem to generate mesodermal or ectodermal derivatives in vitro or in vivo.

5) The manuscript still must be edited by an English native speaker.

Response: The manuscript has been largely overhauled by a native speaker.